# Biotic homogenization, lower soil fungal diversity and fewer rare taxa in arable soils across Europe

Samiran Banerjee [1,2] ✉, Cheng Zhao [3], Gina Garland[2], Anna Edlinger [2,4], Pablo García-Palacios [5,6], Sana Romdhane[7], Florine Degrune[8], David S. Pescador [9,10], Chantal Herzog [2], Lennel A. Camuy-Velez[1], Jordi Bascompte [11], Sara Hallin [12], Laurent Philippot[7], Fernando T. Maestre [13,14], Matthias C. Rillig [8,15] & Marcel G. A. van der Heijden [2,6] ✉

Soil fungi are a key constituent of global biodiversity and play a pivotal role in agroecosystems. How arable farming affects soil fungal biogeography and whether it has a disproportional impact on rare taxa is poorly understood. Here, we used the high-resolution PacBio Sequel targeting the entire ITS region to investigate the distribution of soil fungi in 217 sites across a 3000 km gradient in Europe. We found a consistently lower diversity of fungi in arable lands than grasslands, with geographic locations significantly impacting fungal community structures. Prevalent fungal groups became even more abundant, whereas rare groups became fewer or absent in arable lands, suggesting a biotic homogenization due to arable farming. The rare fungal groups were narrowly distributed and more common in grasslands. Our findings suggest that rare soil fungi are disproportionally affected by arable farming, and sustainable farming practices should protect rare taxa and the ecosystem services they support.

Soil fungi play a crucial role in agroecosystems by delivering essential functions such as mineralization of organic matter to plant-available nutrients, aggregate stability, carbon stabilization, and plant growth promotion[1–4]. However, there is often a trade-off between agricultural production and biodiversity, i.e., intensive agriculture can disrupt belowground communities and result in biodiversity loss[5,6]. Studies performed at local scales have found a negative impact of intensive farming on the richness of specific fungal groups such as arbuscular

[1]Department of Microbiological Sciences, North Dakota State University, Fargo, ND 58102, USA. [2]Agroscope, Plant-Soil Interactions Group, 8046 Zurich, Switzerland. [3]ETH Zurich, Institute for Environmental Decisions, 8092 Zurich, Switzerland. [4]Wageningen Environmental Research, Wageningen University & Research, Droevendaalsesteeg 3, 6708 PB Wageningen, The Netherlands. [5]Instituto de Ciencias Agrarias, Consejo Superior de Investigaciones Científicas, 28006 Madrid, Spain. [6]University of Zurich, Department of Plant and Microbial Biology, 8057 Zurich, Switzerland. [7]University Bourgogne Franche Comte, INRAE, Institut Agro Dijon, Agroecologie, Dijon, France. [8]Freie Universität Berlin, Institute of Biology, Altensteinstr. 6, 14195 Berlin, Germany. [9]Departamento de Farmacología, Farmacognosia y Botánica, Facultad de Farmacia, Universidad Complutense de Madrid, 28940 Madrid, Spain. [10]Departamento de Biología y Geología, Física y Química Inorgánica, Universidad Rey Juan Carlos, 28933 Móstoles, Spain. [11]University of Zurich, Department of Evolutionary Biology and Environmental Studies, 8057 Zurich, Switzerland. [12]Swedish University of Agricultural Sciences, Department of Forest Mycology and Plant Pathology, Box 7026, 750 07 Uppsala, Sweden. [13]Departamento de Ecología, Universidad de Alicante, Carretera de San Vicente del Raspeig s/n, 03690 San Vicente del Raspeig, Alicante, Spain. [14]Instituto Multidisciplinar para el Estudio del Medio "Ramón Margalef", Universidad de Alicante, Carretera de San Vicente del Raspeig s/n, 03690 San Vicente, del Raspeig, Alicante, Spain. [15]Berlin-Brandenburg Institute of Advanced Biodiversity Research (BBIB), 14195 Berlin, Germany. ✉e-mail: samiran.banerjee09@gmail.com; marcel.vanderheijden@agroscope.admin.ch

mycorrhizal fungi[4,7,8]. However, the effect of arable farming on the overall soil fungal communities should also be determined at large scales. Owing to the difference in the levels of resources and inputs in ecosystems (e.g., arable lands vs. grasslands), the response of different functional groups to land-use change might also vary. The impact of production systems on biodiversity is not dichotomous, and land use effects on ecological communities can be nuanced and require large-scale investigations[9,10]. Assessing such effects of land-use intensification on soil fungal biogeography across a wide range of climatic and soil conditions and understanding its consequences for agroecosystem functioning is important. Such information can be used for designing sustainable farming practices that support fungi-mediated ecosystem services and bolster food security. Consequently, there has been a surge of fungal biogeography studies at large scales[4,8,11–14], revealing groups of fungi that are rare or restricted to a few sites. Beyond investigating the overall distribution, a fundamental goal of microbial biogeography studies is to identify the environmental factors that shape microbial distribution patterns[15–18]. Identifying such factors can also reveal why some fungal groups display environmental filtering or provincialism while others are more widespread[1,19], helping develop models on soil fungal responses to land-use intensification at large scales.

The effect of agricultural practices may differ across prevalent and rare taxa[20]. For example, while prevalent groups may not show any noticeable response to intensive practices, rare taxa, groups with a narrow niche breadth[1,12,15], may undergo considerable distributional changes and be present at specific locations or be exclusive to a particular practice. While the influence of anthropogenic activities on plant rarity has been studied for decades[21], little is known regarding their effects on soil microbial communities. This is of particular concern since numerically rare microbial taxa have been shown to play a key role in many soil biogeochemical processes including nitrification, denitrification and methanogenesis[22–24]. Rare taxa are an important constituent of the soil biodiversity reservoir, and their contribution is critical to sustainable ecosystem functioning. In the case of fungi, multiple comprehensive studies have shown that most fungal species are not cosmopolitan and that over 80% of the taxa are rare and likely vulnerable to extinction[8,21]. Intensive management practices may exert a strong homogenizing effect on fungal communities, reducing the occurrence of rare taxa[5,25]. Such biotic homogenization would result in a lower fungal diversity in intensively managed ecosystems[6,14] with implications for fungi-mediated processes. In contrast, rare taxa may be more common in grasslands because of less disturbance and/or the availability of more niches and heterogeneous resources due to greater aboveground diversity in grasslands, especially at a large geographic range. However, the impact of anthropogenic land-use intensification on fungal biogeography and rare taxa is still unclear.

Here, we aimed to address the following research questions: (a) How do fungal diversity and community composition vary across land use and countries? (b) What factors drive fungal biogeographical patterns at continental scale? and (c) Are abundant and rare fungal groups differentially affected by land use? To answer these questions, we used the high-resolution PacBio Single Molecule, Real-Time (SMRT) Sequel sequencing targeting the entire internal transcribed spacer (ITS) region and assessed fungal biogeographical patterns in 156 arable and 61 grassland sites across a 3000 km North-South gradient in Europe (Fig. S1). By incorporating climatic, management, and soil properties, we identified the drivers of soil fungal biogeography. We assessed the prevalence and rarity of soil fungi and how arable farming specifically influences rare fungal groups. Rarity can be defined in different ways to identify taxa that are permanently rare, conditionally rare, and transiently rare[22,26]. In this study, we identified rare taxa by selecting the OTUs present in two or fewer sites. Finally, we assessed the importance of rare taxa for soil ecosystem processes to elucidate whether the loss of such groups can have any functional implications.

We hypothesized that the effect of land use differs between abundant and rare fungal groups. Based on our findings from the root mycobiome[27], we also hypothesized that agricultural intensity has a negative influence on the overall soil fungal diversity.

## Results

### Fungal diversity and composition

Arable lands comprised significantly (Kruskal-Wallis chi-squared = 36.016; $P < 0.001$) less diverse soil fungal communities compared to grasslands across five countries (Kruskal-Wallis chi-squared = 48.86; $P < 0.001$) in Europe (Figs. 1A, S2, S3). This difference was consistent when fungal richness was computed for the same number of samples for grasslands and arable lands (Fig. S4; $n = 61$). Overall, on average, arable lands harbored 20% fewer OTUs than grasslands (Table S1). Geographic proximity exerted a significant effect (PERMANOVA $F = 14.369$; $R^2 = 0.206$; $P < 0.001$) on fungal communities with sites clustering based on their distribution from Sweden in the north to Spain in the south (Figs. 1B, S5, S6). Within each cluster, the arable lands and grasslands also tended to group significantly (PERMANOVA $F = 11.369$; $R^2 = 0.040$; $P < 0.001$). The interactive effect of country and land use was also significant (PERMANOVA $F = 2.335$; $R^2 = 0.032$; $P < 0.001$). Major fungal classes also showed significant differences between land uses (Fig. S7; Table S2 and S3). In particular, Sordariomycetes, Mortierellomycetes and Tremellomycetes were relatively more abundant in arable lands than in grasslands (Fig. 1C). Several functional groups of soil fungi also varied significantly across the five countries (Fig. 1D). Arbuscular mycorrhizal fungal groups (Glomerallales; Table S2) showed a higher relative abundance in grasslands than in arable lands, which may be due to the availability of host plants in grasslands.

### Factors influencing fungal biogeography

The prevalent fungal classes showed distinct distribution patterns across the latitudinal gradient (Fig. S8). There were also positive associations between fungal richness and latitude and longitude, with the highest richness in mid-latitude (Fig. 2A–D). However, the associations were stronger for grasslands than arable lands. Although Switzerland had the highest richness among all countries, the relationships still held true when assessed without the Swiss samples (Fig. S9). Edaphic factors were an important driver of fungal richness in arable lands (Fig. 2C), with soil pH, carbon, nitrogen, and cation exchange capacity having the strongest correlations with fungal diversity across both arable lands and grasslands (Figs. S10, S11, S12). Climatic factors also displayed consistently strong associations in both arable and grasslands across the continent (Fig. 2D). Partial Mantel tests revealed that environmental parameters had a stronger influence in grasslands ($r_{Env|Geo} = 0.239$, $p < 0.001$) than in arable lands ($r_{Env|Geo} = 0.112$, $p = 0.008$). Similarly, when accounted for the geographic distance, a stronger effect of climate was observed for grasslands ($r_{Clim|Geo} = 0.141$, $p = 0.006$) than arable lands ($r_{Clim|Geo} = 0.03$, $P = 0.23$). Effects of soil properties were also significant in both arable ($r_{Soil|Geo} = 0.228$, $p = 0.001$) and grassland ($r_{Soil|Geo} = 0.243$, $p = 0.002$) sites. We further explored these factors through structural equation modeling (SEM), which ($\chi^2$ test $p_{arable} = 0.294$, $p_{grasslands} = 0.699$) explained a larger proportion of variance in fungal richness in grasslands ($R^2_{grasslands} = 40.8\%$) than in arable lands ($R^2_{arable} = 24.7\%$) (Fig. S13). Nitrogen mineralization potential and mean annual temperature were the most important drivers of fungal diversity in grasslands while soil carbon content was the main driver in arable lands.

### Soil fungal rarity

Grasslands had a proportionally higher presence of rare taxa compared to arable lands (Fig. 3). Cumulative abundance was calculated by square-rooting the number of reads per sample, and then averaging and summing up the ordered relative abundances. The cumulative

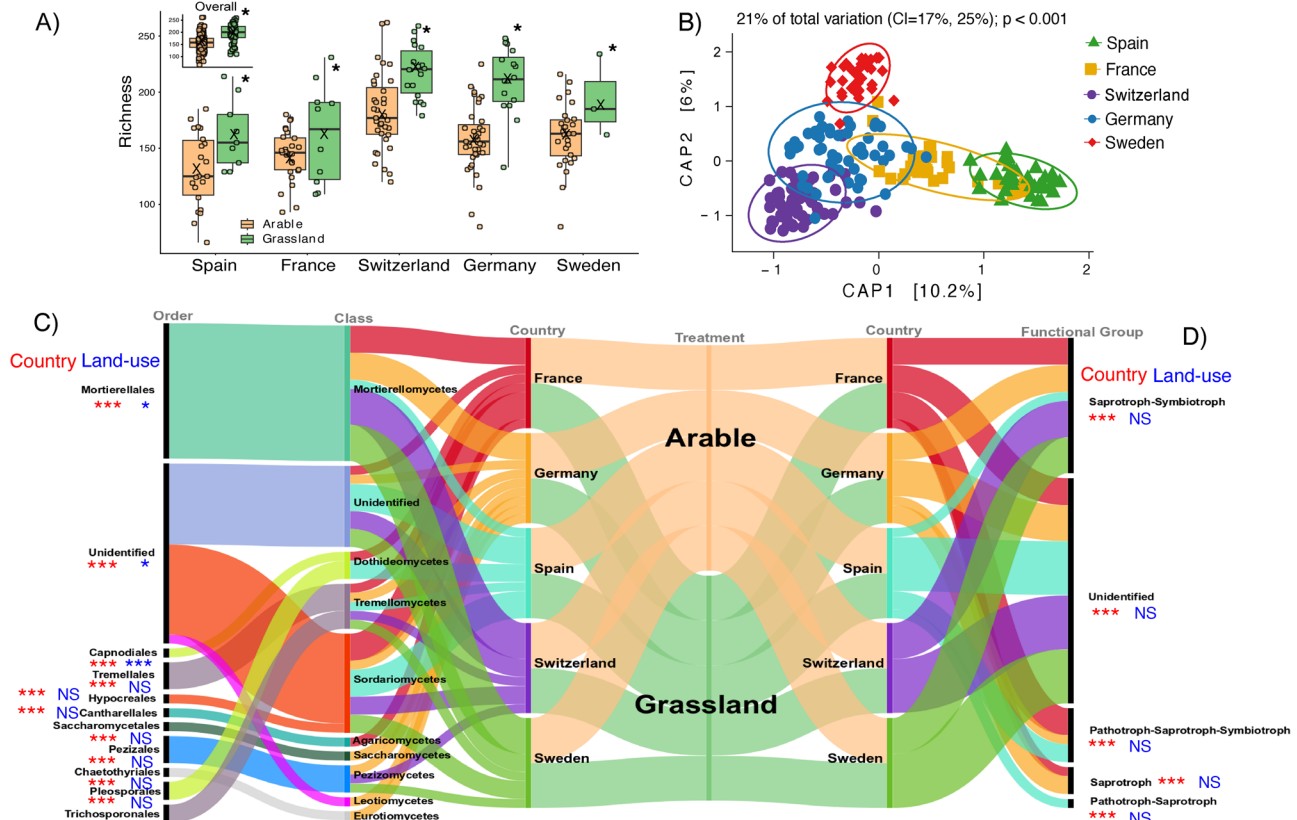

**Fig. 1 | Diversity, structure, and composition of the soil mycobiome. A** Fungal richness in arable lands ($n = 156$) and grasslands ($n = 61$) across five countries in Europe. Countries are arranged from the south on the left side to the north on the right side. For each country, the left (orange) and right (green) boxes indicate richness in arable lands and grasslands, respectively. The overall richness of soil fungi is shown in the upper-left subplot. * indicates significant ($P < 0.05$) difference between arable lands and grasslands. Small circles indicate individual data points, boxes mark the interquartile range, vertical lines indicate the whiskers, bold horizontal lines show the median and 'x' indicates the mean value. **B** Principal Coordinate Analysis (PCoA) showing Bray-Curtis dissimilarity of soil fungi in arable lands and grasslands across five countries. Colors represent five European countries.

Alluvial diagrams showing the relative abundance of the abundant classes and orders (**C**) of soil fungi and the potential functional groups (**D**) in arable lands and grasslands across five counties. In these diagrams, various blocks represent clusters, and the streams or flows represent changes in the composition. For each country, the height of the blocks represents the size of cluster of fungal groups. Relative abundance of a functional group was determined by summing up relative abundances of OTUs that belonged to that specific group. The effects of geographic location (in red color) and land use type (in blue color) as assessed by Kruskal Wallis rank sum test are shown on the sides. Statistical significance: *$p$ value < 0.05; **$p$ value < 0.01; ***$p$ value < 0.001; n.s not significant.

abundance was similar for the most abundant OTUs, but the difference increased for the rare (lower ranked) OTUs, with the square-rooting of the total number of sequences per site reaching 707.4 for grasslands compared with 589.4 for arable lands (Fig. 3A). Arable lands had 16.7% fewer fungal OTUs than grasslands and this was largely due to fewer rare OTUs. Furthermore, 75% of the total reads in arable lands was made up of the 19 most prevalent OTUs, while it required the 30 most prevalent OTUs to achieve 75% reads in grasslands. The cumulative abundance slopes for the two land use types were similar near the origin due to the prevalent OTUs, but they diverged due to fewer less abundant OTUs in the arable lands (Fig. 3A), and this pattern was consistent across all countries (Fig. S14). There was a positive correlation between OTU abundance and site occupancy or commonness (the number of sites the OTUs were present at) with rare OTUs significantly ($t$-test $p < 0.05$) more common in grasslands (Fig. 3B). Agricultural intensity, calculated[28] based on the tillage intensity and the application of chemical fertilizers and pesticides, was negatively associated with both the overall fungal richness ($\rho = -0.27$; $P < 0.001$) as well as the richness of rare taxa ($\rho = -0.24$; $P < 0.004$). Consistent with our hypothesis, these results suggest a possible negative impact of agricultural intensification on the soil mycobiome (Figs. 3C; S16).

On average, 40.7% of all OTUs were present in both land use types, whereas 19.5% and 39.8% were specific to arable lands and grasslands,

respectively (Table S1). In line with our hypothesis, the richness of rare fungi was significantly ($P < 0.001$) lower in arable soils, and this was similar when compared with the same number ($n = 61$) of samples of both land-use types (Figs. 4A, S15). The total number of rare OTUs was also higher in grassland soils (535) compared to arable soils (331). Although Spain had the lowest fungal richness, its proportion of rare taxa (0.154) was considerably higher than that of the other countries, especially France (0.073) and Sweden (0.076), suggesting that rarity was not proportional to the overall diversity. Several orders of the rare taxa were more common in grasslands than arable lands, including Pleosporales, Thelephorales, Pezizales, and Hypocreales, which are known for their responsiveness to plant diversity (Fig. 4B). While a large majority of rare fungi were unclassified at the genus level, we found a noticeable presence of the members of Mortierella, Archaeorhizomyces, Entoloma, Pluteus, and Psathyrella genera. Arbuscular mycorrhizal fungi of the order Glomerales were also significantly ($P < 0.001$) more abundant in grasslands, while plant pathogens were relatively more abundant in arable lands (Fig. 4B). We also annotated the ITS sequences to KEGG functional categories to identify the molecular functions of rare fungal taxa (Fig. S16). The major functions were associated with metabolism of carbohydrates (25%), amino acids (22%), and lipids (10%). Another important functional pathway was the biosynthesis and metabolism of glycans, which are key constituents of

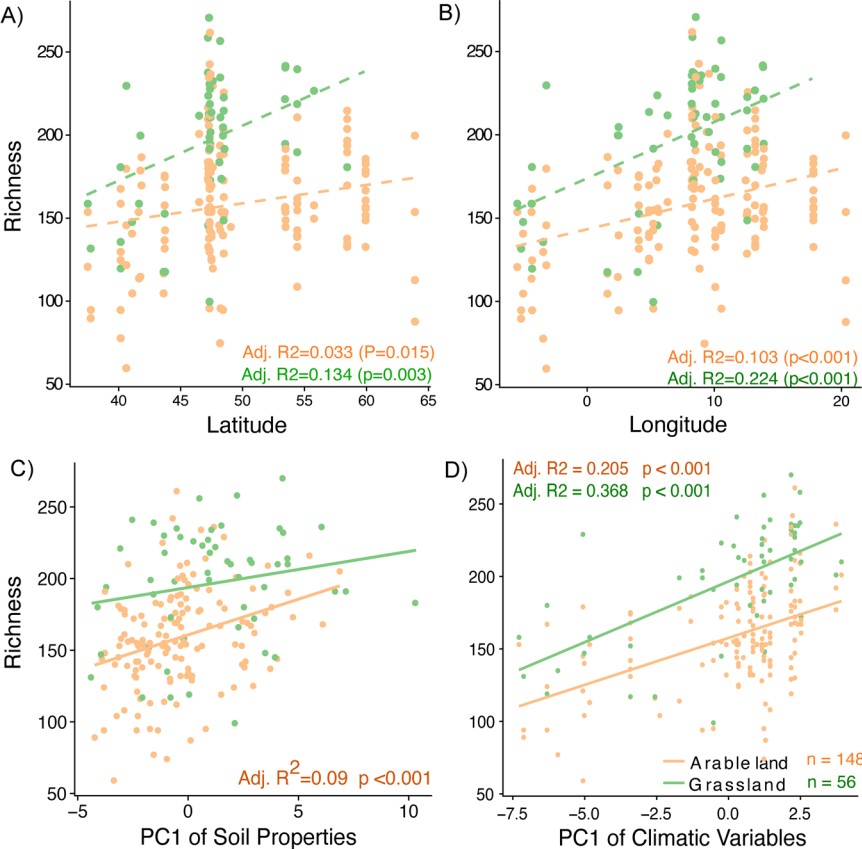

**Fig. 2 | Drivers of fungal biogeography. A**, **B** Latitudinal and longitudinal gradient of fungi richness in arable lands (orange) and in grasslands (green). First order polynomial models show that strong effects of latitude and longitude on fungal richness in arable lands and grasslands. **C** Relationship between fungal richness and the principal component (PC) 1 of soil properties and soil fungal richness. The relationship was only significant in arable lands. **D** Relationship between fungal richness and the principal component 1 of bioclimatic variables. Fungal richness increased significantly with increasing PC1 in both land use types. Adjusted R2 and $p$ values indicate statistical significance.

the fungal cell wall. Rare soil fungi were also positively and significantly correlated with important soil ecosystem functions across both land-use types (Fig. S17), indicating that the loss of such groups due to agricultural intensity might have functional implications. However, since these groups were a part of the rare mycobiome, the correlations were weak. Overall, our results show a similarity between relative abundance and commonness, with abundant groups being more common, while rare groups even rarer.

## Discussion

We used the high-resolution PacBio Sequel targeting the entire ITS region to dissect fungal biogeographical patterns in 156 arable sites and 61 extensively managed grasslands across a 3000 km North-South gradient in Europe. We show that fungal diversity is consistently lower in arable lands than in grasslands across five countries. Rare fungi were affected or absent in arable lands, suggesting that biotic homogenization and a disproportionally negative impact of arable farming on the rare soil microorganisms.

### Biotic homogenization in arable lands

We found that, on an average, arable lands across the five European countries had nearly 25% lower fungal diversity than grasslands. One of the major impacts of intensive agriculture is the loss of biodiversity[5,6]. Practices such as excessive use of synthetic fertilizers and pesticides, monocultures, excessive tillage, and the homogenization of landscapes can negatively affect the local and regional pool of biodiversity[5,29]. Indeed, agricultural intensification has been linked to a systematic decline in birds, invertebrates and amphibians in arable lands[6,30]. Importantly, biodiversity loss is not just a decrease in species number, but it also accompanies the loss of associations among various species, with the potential disruption of the network of mutual dependencies between species. For example, our previous report found that agricultural intensification has a negative impact on root endophytic fungi, and the associations among fungal members in conventional farmlands is 50% less than in organic lands[27]. Recent studies also found that the abundance of beneficial mycorrhizal fungi is negatively associated with pesticide residues[31] and pesticide application reduces the richness of mycorrhizal fungi and their ability to acquire phosphorus from the soil[32]. Despite this, we have limited knowledge of how intensive agricultural practices affect soil fungal diversity and distribution at large scales[33]. Here, we show that fungal diversity peaked at mid-latitude, although it was consistently lower in arable lands. There was a negative relationship between fungal diversity and the agricultural management intensity calculated from tillage, agrochemicals, and pesticide information (Fig. 3C). This is in contrast to a recent study which found that fungal diversity was similar in grasslands and non-permanent croplands[14], which could be due to the wider range of crops (over 20 different crops including cereals and vegetables) sampled in that study. Most of the arable lands assessed in this study practiced conventional tillage, which has been found to negatively affect mycelial networks[34], and may be a cause of the lower fungal diversity in arable soils. Furthermore, fertilization applications in arable lands can cause resource homogeneity, which may result in the dominance of copiotrophic microorganisms. Arable systems also have fewer hosts for symbiotic groups. Extensively managed grasslands are an important type of agricultural system that covers nearly

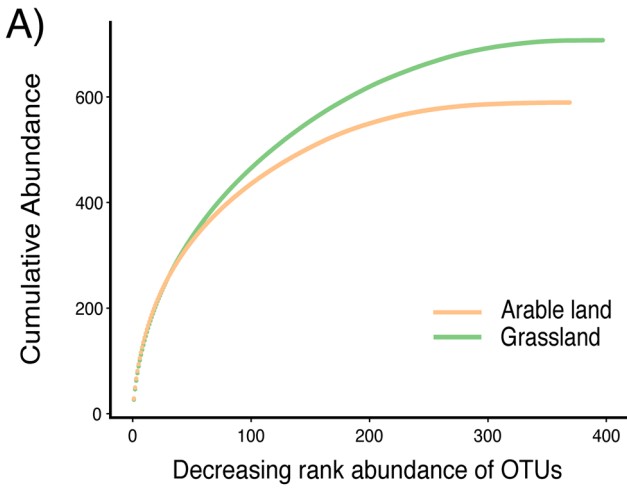

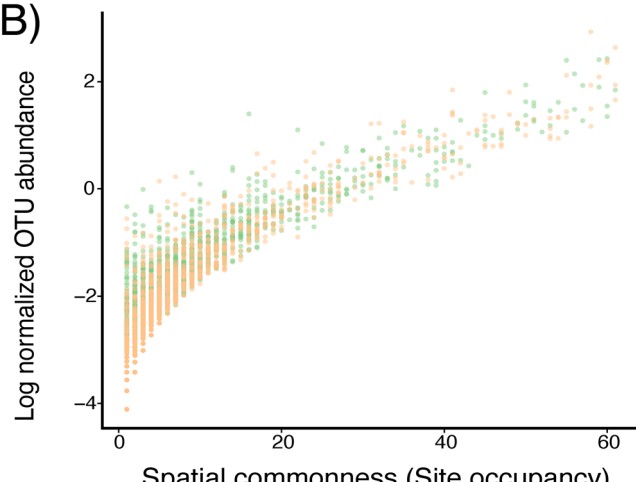

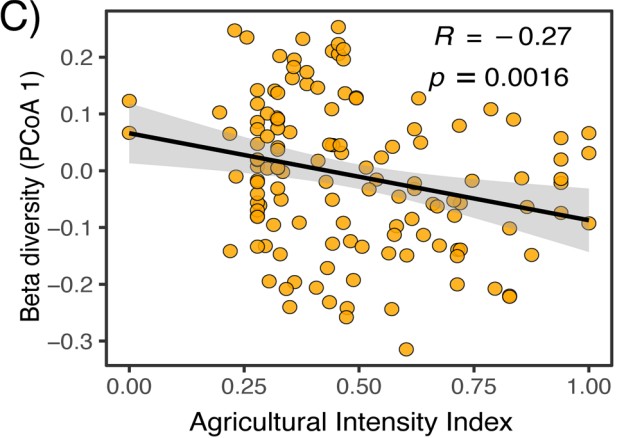

**Fig. 3 | Spatial commonness and rare fungi. A** The relationship between abundance and commonness of the overall soil fungal communities in arable lands ($n = 156$; orange) and grasslands ($n = 61$; green) across all five countries. Abundance is represented by the cumulative abundance of OTUs, which was calculated as the square-root transformed number of reads per sample. Cumulative abundance curves were created by averaging and summing up the ordered abundances. Commonness represents decreasing rank abundance of OTUs. Difference between the two land use types was mainly due to rare taxa, i.e., while abundant OTUs (higher ranked on the left) are similarly abundant in both land-use types, grasslands had a higher number of rare OTUs (lower ranked OTUs towards the end of the curves). **B** Spatial commonness or site occupancy of the overall soil fungal communities as revealed by the abundance of OTUs and the number of sites they were present at. Commonness represents the number of sites that the OTUs were present at, and it was calculated by square-root transforming the OTU abundance and then log-transforming to plot against the number of sites occupied. Common OTUs were present in a higher number of sites while rare OTUs were restricted in fewer sites. **C** Relationship between the overall fungal diversity and agricultural intensity for arable sites ($n = 156$). The Y axis represents the principal coordinate 1 of the overall OTUs using Bray-Curtis similarity whereas the X axis represents the agricultural intensity index calculated from agrochemical applications, tillage intensity, and crop diversity. Error bands indicate 95% confidence interval.

diversity could not be tested in our study. Moreover, we sampled extensive grasslands, which are different from pristine native grasslands in terms of their species composition. As shown in previous studies[37], plant diversity can be an important determinant of soil fungi as a diverse plant community influences soil physical properties with different root architectures, shapes the soil chemistry with diverse root exudates and residues, and thereby modulates the soil mycobiome. For example, labile resources will attract copiotrophic fungi while oligotrophic groups will settle for more recalcitrant resources. Indeed, resource heterogeneity may also vary based on the crop rotations in arable lands[38]. Future studies should measure plant diversity when assessing arable and native ecosystems to dissect above- and belowground linkages.

### Drivers of fungal biogeography
Understanding the drivers of microbial distribution patterns is a fundamental goal of microbial ecology. Even in the global surveys of soil fungi[4], agricultural systems are often not considered, and thus, our knowledge of the drivers of soil fungal biogeography in agricultural systems is still rudimentary. A large-scale investigation comparing fungal biogeography across intensive agricultural and extensive grassland systems can reveal the relative importance of climatic, geographical, and edaphic factors, and whether some drivers are consistent across intensive and extensive land-use types. Grassland and arable sites were paired in this study, making direct comparisons easier. We found that fungal diversity increased with latitude and longitude, with the highest diversity at mid-latitude (45°–50°). Importantly, this positive association was stronger in grassland soils, which is congruent with previous global reports on natural soils[4,11]. Geographic distance and environment had a stronger influence in grasslands than in arable lands. The importance of spatial distance and soil environmental properties for structuring fungal communities has also been observed previously in non-agricultural soils[12,39]. Owing to the inherent heterogeneity in soil resources, fungi communities vary across space and understanding of this community turnover can yield insights into the factors that govern their environmental filtering and dispersal limitations[13]. Climatic factors emerged as a strong control on soil fungi when geographic distance was accounted for. Previous reports on non-agricultural soils found a positive relationship between fungal diversity and mean annual temperature, with fungal diversity decreasing across the continent[4,11]. In this study, both mean annual precipitation (366 mm–1296 mm) and temperature (2.8 °C–17.9 °C) varied considerably across the continent, and we found that fungal communities were more diverse in areas at mid-latitude (45°–50°) that

16% of all lands in Europe[35]. Farmers are encouraged and often required to maintain a certain proportion of their lands as grasslands in order to receive government subsidies[36]. The consistently higher fungal diversity in grasslands supports the maintenance of grasslands as part of the farmlands to promote the soil mycobiome. We also found a greater abundance of mycorrhizal fungi and a relatively lower abundance of pathogens in grasslands. Overall, our results show that soil mycobiome displays contrasting biogeographical patterns between land-use types but a consistency across the 3000-km European gradient. However, plant community composition was not measured for grassland sites in this study and as a result, the importance of plant

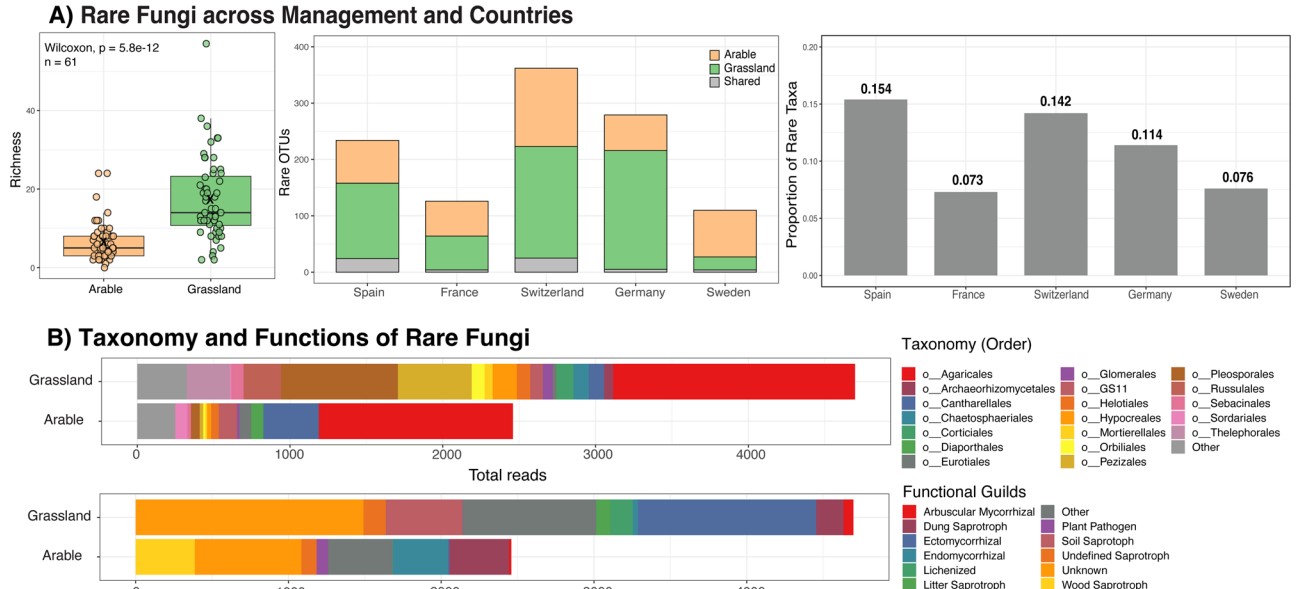

**Fig. 4 | Proportion and composition of rare fungi. A** Rare fungal richness and proportion across land-use types and countries. Wilcoxon rank sum test was performed on the same number of sites (*n* = 61) for each land use. Left Panel: The richness of rare fungi per sample was significantly higher in grassland soils when compared with the same number of samples from arable lands. Small circles indicate individual data points, boxes mark the interquartile range, vertical lines indicate the whiskers, bold horizontal lines show the median and 'x' indicates the mean value. Middle Panel: The cumulative number of rare OTUs were also higher in grassland soils. Right Panel: Proportion of rare taxa was calculated by dividing the number of rare fungi with the total number of OTUs. **B** Taxonomic (upper panel) and functional (lower panel) composition of rare taxa in arable vs grassland soils. Rare taxa were identified from the original OTU table by identifying OTUs that are only present in one or two sites at respective land use types and Levin's niche breadth less than 0.55.

had low temperature but high precipitation. Indeed, higher temperatures reduced fungal diversity in both land-use types and fungal diversity was higher when the temperature change between seasons was more gradual. Collectively, these results show a consistent effect of geographic location and land-use on soil fungal communities across European countries.

### Fungal rarity and possible implications

The distribution and drivers of soil fungal biogeography may vary between agricultural systems that routinely receive amendments and extensively managed systems that do not. Thus, a large-scale investigation comparing fungal biogeography across agricultural and native systems would reveal if intensive agricultural practices can result in location-exclusivity of fungi, and whether such patterns are consistent across large spatial scales. Rare and abundant fungi can vary in their responses[20]. Here, we found a disproportional effect of agricultural intensification on soil fungi in which prevalent taxa were unaffected, while rare fungi were rarer in arable lands compared to extensively managed grasslands. Consistently, agricultural intensity was negatively linked to the richness of both overall fungi and rare groups. Our observations are consistent with studies on macroorganisms, which report that rare species are vulnerable to anthropogenic disturbances and are at greater risk of extinction[40,41]. While there is a strong emphasis on dominant taxa in recent studies[37,38], our results suggest that the role of rare fungal taxa in soil ecosystems must not be overlooked. This is important because previous studies have found over 80% of fungi are rare or endemic and less than 100 groups are cosmopolitan[1,12], and thus, rare members constitute a critical portion of belowground diversity. Although the overall richness was the lowest in Spain, it had the highest proportion of rare fungi, which might be pointing towards a greater vulnerability of rare fungi in dryland ecosystems. Indeed, a recent study found that fungi are vulnerable to drought at the global scale[8]. However, further studies are needed to compare such patterns in both dryland- and non-dryland ecosystems to obtain robust conclusions.

Rare taxa have been shown to play a key role in many soil biogeochemical processes including nitrification, denitrification and methanogenesis[22–24,26]. In this study, we also found rare taxa were significantly associated with important soil processes such as basal respiration and aggregate stability, which are important indicators of soil health[42]. Thus, the loss of rare taxa might have functional implications for ecosystems. Conservation efforts must be made to protect such taxa and maintain their contributions to ecosystem functioning. For example, rare fungi display unique signatures compared to the abundant taxa in endangered Antarctic ecosystems[43]. Rare members might be critically important for nutrient cycling in vulnerable ecosystems, and they might be indispensable for the survival of rare or endemic plants, hypotheses that can be tested in the future.

It is important to note that rare taxa can be dormant microbial members or a part of the relic DNA pool in the soil. Future studies may wish to identify active rare taxa by treating DNA samples with propidium monoazide (PMA), which binds to relic DNA and prevents subsequent amplification[44]. It would also be of interest to see the relative contributions of eco-evolutionary factors to fungal rarity in agricultural systems. For example, an interesting question would be to explore if dispersal limitation exacerbates rare fungal distribution in arable lands. Future studies may also investigate how the loss of rare fungi affects functional plasticity and functional contingency of agroecosystems. Lastly, fungal biogeographical distribution can display temporal variability[13], and thus, future studies should also assess fungal diversity over seasons or years to understand the temporal nature of biogeographical patterns. Temporal patterns of soil fungi would reveal if different types of rare taxa (e.g., permanent, transient, and conditional) are also disproportionally affected in arable lands.

### Concluding remarks

Intensive agriculture is a major cause of the loss of soil biodiversity of which soil fungi are an important constituent. Here we report that soil fungi display distinct biogeographical patterns at the continental scale with consistently lower diversity in arable lands than grasslands. We

then show that arable farming leads to a homogenization of fungal communities with prevalence of abundant groups in arable soils while rare community members are rarer or even absent. We also report an adverse effect of agricultural intensity on the richness of overall fungi as well as rare groups. Our observation on soil fungi is consistent with studies on macroorganisms reporting that rare species are particularly susceptible to anthropogenic disturbances and at greater risk of extinction[45,46]. This is important because global biodiversity conservation efforts recognize the vulnerability and irreplaceability of rare and endemic plants and animals, but largely overlook soil biodiversity[47]. Further, many of the extinction factors have been shown to act synergistically, with exacerbating effects on rare species[48]. Soil is one of the largest reservoirs of biodiversity[49,50] and rare species are a key constituent of that[1]. Similar to plant- and animal hotspots, specific ecoregions should also be identified for rare microbiota. One of such ecoregions might be the dryland agroecosystems, where we found a higher proportion of rare fungi, however, further studies are necessary to understand the spatiotemporal variability and extinction dynamics of such groups.

## Methods

### Site selection and soil sampling
This study was a part of the Digging Deeper Project conducted across five European countries Sweden, Germany, Switzerland, France, and Spain (Fig. 1). A total of 217 agricultural fields were chosen, including 156 arable sites and 61 extensively managed grasslands[28,51]. A majority (78%) of the arable sites were planted with wheat (*Triticum aestivum*) ($n = 121$) with the other cereal crops such as barley, *Hordeum vulgare* ($n = 26$); oat, *Avena sativa* (n = 6); rye, *Secale cereale* ($n = 1$); or triticale, Triticosecale sp. ($n = 1$) selected when wheat was unavailable. When possible, we paired agricultural fields with non-arable lands by sampling nearby extensively managed grasslands and marginal lands with permanent, predominantly herbaceous plant cover. These non-arable sites were mostly unfertilized and occasionally mowed. Many of these arable lands did not have nearby extensively managed grasslands, which resulted in an unbalanced design. However, to address whether the unbalanced design affected the outcome of this study, we computed fungal richness on the same number of arable lands ($n = 61$) by randomly selecting fields. Our analysis revealed that fungal richness was still lower for arable lands than grasslands (Fig. S4). Soil samples were collected in Spring 2017. At each site, eight soil cores were obtained in a circular pattern within a 10 m radius using a 5 cm diameter step-probe and to a depth of 20 cm. Soil samples were kept on ice until their transfer to the laboratory. Three of the cores were kept intact and used to measure bulk density and soil aggregation. The remaining soil cores were homogenized and sieved to 2 mm. Soil subsamples were air-dried for further processing for soil physical and chemical properties, stored at 4 °C for soil properties such as microbial biomass, and frozen at −18 °C for DNA extraction, mineral nitrogen content and potential N cycling rates.

### Soil analyses
Soil properties were assessed according to Swiss Standard Protocols[32,52] and conducted in Agroscope Reckenholz, Zurich. Gravimetric moisture was determined with a 10 g of field-moist soil at 105 °C for 24 h. Soil pH was determined with a 20 g of soil in a 1:2.5 soil:water solution. Soil texture was determined following the hydrometer method[53]. Mineral nitrogen content ($NH_4^+$-N and $NO_3^-$-N) was determined by extracting soil samples with 1 M KCl (soil: solution ratio of 1:10) and analyzed on a San++ Automated Wet Chemistry Analyzer-Continuous Flow Analyzer (CFA, Skalar, The Netherlands). Cation exchange capacity was estimated with a 2.5 g of soil and the results were expressed in mg/L. Total carbon and total nitrogen were analyzed by combustion of a 250 mg of soil on a TruSpec CN Analyzer (LECO, MI, USA). Soil organic carbon content was analyzed with a 0.5 g of soil

using the potassium-dichromate ($K_2Cr_2O_7$) oxidation method. Microbial biomass carbon and microbial biomass nitrogen were measured by the chloroform fumigation method[54]. Chloroform fumigation was performed with triplicates of 20 g of soil samples incubated for 24 hours. Basal respiration was measured by incubating soils with a NaOH solution for 24 hours. Soil carbon and nitrogen concentration were determined by combusting a 250 mg of soil using TruSpec CN Analyzer (LECO, MI, USA).

### Amplicon sequencing
Soil DNA was extracted from 0.25 g soil using the PowerSoil DNA isolation kit (MoBio, Carlsbad, CA) following the manufacturer's instructions. Fungal ITS region was amplified using the PacBio SMRT Sequel platform with the primers ITS1F (CTTGGTCATTTAGAGGAAGTAA) and ITS4 (TCCTCCGCTTATTGATATGC) targeting the entire ITS region[55]. A two-step PCR was conducted on a Biorad PCR Instrument (Biorad, Hamburg, Germany) using the 5PRIME HotMaster Taq DNA Polymerase (Quantabio, Beverly, MA, USA) in 20 μl of reaction mixture[56]. Details on PCR conditions and library preparation can be found in the Supplementary Information. The sequencing libraries were prepared using P6/C4 chemistry (DNA/Polymerase Binding Kit P6, DNA Sequencing Reagent 4.0) on the PacBio® Sequel Instrument available at the Functional Genomic Centre Zurich (FGCZ, Zurich, Switzerland; http://www.fgcz.ch). The PacBio SMRT Portal (https://www.pacb.com/products-and-services/analytical-software/smrt-analysis/) was used to process the raw sequences and extract the circular consensus sequences of at least five passes[57].

Fastq files obtained from the PacBio runs were quality filtered using the PRINSEQ-lite v0.20.4[58]. Filtering parameters were: GC range 30–70, minimum mean quality score of 20, no ambiguous nucleotides, low sequence complexity filter with a threshold of 30 in the DUST algorithm. In a next step, the reads were demultiplexed using an in-silico PCR approach as part of USEARCH v11[59] allowing max 1 mismatch in the barcode-primer sequence but not at the 3-prime ends. The amplicon size range was set to 100–2000 but all amplicons with additional primer sites (concatenated amplicons - multi-primer artefacts) were removed. Sequences were then clustered into operational taxonomic units (OTUs), based on 97% similarity, using the UPARSE pipeline (Edgar 2013). Taxonomical information was predicted to OTUs based on the UNITE database (V7.2)[60] using the SINTAX classifier[61]. Singletons (OTUs that only occur once) and OTUs with low abundance (relative abundance less than 0.5% in total and less than 0.5% in each sample) were excluded. A total of 17 phyla, 42 classes, 90 orders, 187 families and 293 genera were classified. Only 6% OTUs were identified at the species level. Nearly half (49.4%) of the OTUs were classified to at least the class level, accounting for 80.2% of total abundance. The OTU table was rarefied to 2000 reads per sample (Fig. S2).

### Statistical analyses
All statistical analyses were conducted using packages in R (v.3.4.3). Climatic data were acquired at a resolution of 10 min (around 18.5 km at latitude 40°) including 19 bioclimatic variables (http://www.worldclim.org)[62]. Details on climate data acquisition can be found in the Supplementary Information. Alpha diversity indices such as richness (number of OTUs), Shannon-Wiener index and ACE index were calculated using the *phyloseq* package version 1.16.2[63]. We performed a Kruskal-Wallis rank-sum test to assess whether fungal diversity differed significantly between land use types. Information on putative functions was obtained using the *FunGuild* database[64].

Land-use-specific taxa were identified as the OTUs only present in either arable lands or grasslands. To compare fungal relative abundance across two land-use types and five countries, OTU numbers were normalized by square-rooting number of sequence reads and ordered by decreasing relative abundance for each site. Then, cumulative

abundance curves for arable lands and grasslands were generated by averaging and summing up the ordered abundances, and this was conducted for the overall dataset as well as for each country separately. Welch's $t$-test was applied to test for each ordered OTU whether its relative abundances differed significantly between arable lands and grassland. First- and second-order polynomial models were fitted to the longitudinal and latitudinal distribution of soil fungal diversity, and the best fit was selected based on the corrected Akaike Information Criterion (AICc) using the *AICcmodavg* package[65]. Spearman's correlation coefficients between fungal diversity and bioclimatic variables and soil properties were calculated using the *corrplot* package. Since many variables for bioclimatic attributes and soil properties were correlated with each other, principal component analysis (PCA) was conducted, followed by linear regression analysis with the first principal component (PC1) as the explanatory variable and fungal diversity as the dependent variable. If two variables correlated too strongly with each other (Spearman's $r > 0.95$), only one was retained for the PCA analysis. Fungal community structure was assessed by performing the principal coordinate analysis (PCoA) with Bray-Curtis dissimilarity using the *ordinate* function in *phyloseq*. To examine fungal composition across countries, PERMANOVA was conducted with 999 permutations using the *adonis* function in *vegan*[66]. As PERMANOVA results were significant, canonical analysis of principal coordinates analysis (CAP) was conducted as a constrained ordination to quantify the effects of country and land use on soil fungal composition using the *ordinate* function. To investigate how soil fungal communities differed with increasing geographical and environmental dissimilarities, a partial Mantel test was conducted with the *mantel.partial* function in *vegan*. We performed SEM to investigate complex relationships among geographical locations, climatic conditions, soil properties and fungal communities using the *lavaan* package version 0.6–3[67]. An initial SEM was constructed based on the understanding of factors that shape soil fungal distribution (Fig. S18). Briefly, geographic locations influence soil fungal diversity through fungal dispersal limitations. Climate and soil properties are the two environmental filters on soil fungal communities. Texture, isothermality and mean temperature in the wettest season were also removed to achieve better model fit and parsimony. Samples containing missing values were removed, resulting in 143 observations for arable lands and 54 observations for grasslands, with 8 degrees of freedom. The initial SEM was modified sequentially by removing links that were not significant and hindered model fit, or by adding links that had high modification indices. The final SE model had adequate fit ($p_{arable\ lands} = 0.294$; $p_{grasslands} = 0.699$; Fig. S13). A model was considered acceptable when the chi-square test $p$-value was greater than 0.05. The best model was preferentially selected based on parsimony, coefficient of determination of fungal diversity and goodness of fit using the maximum likelihood approach[68].

Rare taxa were identified from the original OTU table by identifying OTUs that are only present at one or two sites of respective land use types and have a narrow niche breadth (<0.55). We computed niche breadth using the MicroNiche package[69], which calculates the proportional occurrence of taxa across sites. A species is considered generalist when it utilizes all resources equally ($B_n = 1$) and the species has a broad, non-discriminatory niche[69]. On the other hand, a species is considered specialist when it has a narrow, discriminatory niche ($B_n < 0.5$). A null distribution with 999 permutations was used to determine the specialist taxa from the generalist. We used the *FunGuild* database[64] to identify the major functional groups that these rare taxa belong to and verified the functional guilds using the FungalTraits database[70]. To assess the cellular and molecular functions of rare fungi, we used FunFun[71], which is a functional annotator that predicts the gene content of individual fungi from ITS sequencing data.

We created a single index of management intensity that could be compared across arable sites and countries, and this index was used in a previous study[28]. For this, we used the management data obtained from the 2017 growing season. The specific management variables that were used in this index were the number of insecticide, herbicide and fungicide applications, the number of tillage events, the maximum tillage depth, and the total amount of mineral nitrogen applied. All of the above parameters were then included in a single management index using PCA and scaling the index between 0 and 1, with 0 indicating the minimum intensity of management practices and 1 indicating the maximum intensity.

### Reporting summary
Further information on research design is available in the Nature Portfolio Reporting Summary linked to this article.

## Data availability
All data are available on GitHub (https://github.com/sambanerjee2022/Agricultural-intensification-and-fungal-rarity.git). Sequences generated in this study are available through Sequence Read Archive (SRA) under BioProject accession number PRJNA1043689.

## Code availability
All scripts are available on GitHub (https://github.com/sambanerjee2022/Agricultural-intensification-and-fungal-rarity.git).

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

## Acknowledgements
We thank all the farmers and farm managers for allowing us to sample their fields and for completing our detailed questionnaires. We also thank A. Saghai, A. Held, A. Bonvicini, S. Müller, S. Zhao, V. Somerville, A. Brugger, O. Scholz, D. Bugmann, R. Heiz, B. Seitz and M. Roser for help with both field work and laboratory analyses. We thank Dr. Jean-Claude Walser at the ETH Zurich Genetic Diversity Center for his help with bioinformatics. The Digging Deeper project was funded through the 2015–2016 BiodivERsA COg call for research proposals, with the national funders Swiss National Science Foundation (grant 31BD30-172462), Deutsche Forschungsgemeinschaft (317895346), Swedish Research Council Formas contract 2016- 0194), Ministerio de Economía y Competitividad (Digging_Deeper, reference PCIN-2016- 028) and Agence Nationale de la Recherche (ANR, France, grant ANR-16-EBI3-0004-01). J.B. is funded by the Swiss National Science Foundation (grant 310030- 197201).

## Author contributions
M.G.A.v.d.H., S.H., F.T.M., L.P. and M.C.R. designed the study and obtained research funding. S.B., G.G., A.E., P.G.P., C.H., D.S.P., S.R. and F.D. contributed to data collection and analysis. S.B., C.Z. and L.A.C.V. conducted data analysis. S.B., C.Z., J.B. and M.G.A.v.d.H. were involved in the interpretation of results. S.B., C.Z. and M.G.A.v.d.H. drafted the manuscript with contributions from all co-authors. All authors commented on and approved the final manuscript.

## Competing interests
The authors declare no competing interests.
