## [Peer Review File · Nature Communications]

Reviewers' Comments:

Reviewer #1:

Remarks to the Author:

The research article 'Agricultural intensification is linked to the rarity of soil fungi across Europe' by Banerjee et al. describes soil fungal community patterns (mainly diversity) along a 3,000 km gradient. The authors compared fungal community diversity and composition between croplands and grasslands, and observed that grasslands are more diverse.

It is my opinion that descriptive articles (particularly of large datasets) deserve to be published, when done right. This article suffers from overlapping and confusing analyses, poor data visualization, extremely deficient statistical reporting, and overinterpretation of results. Honestly, I doubt that all co-authors (leaders in their respective fields) have read and approved to submit this manuscript as it is. I am very sorry to say that the manuscript feels more like a first rough draft with a haphazard collection of data analyses, rather than a well-thought and revised version.

GENERAL COMMENTS

- Let's start with the title and their main result. The main result is about comparing croplands vs grasslands, and it is completely fine. Agricultural intensification only appears in the random forest analyses. You need to assess the collinearity between those variables. In my opinion, it would make much more sense if SEM analyses were used with the subset of cropland samples to answer the question about agricultural intensification (how climate and agricultural intensification both affect soil, and the direct and indirect effects to soil microbial communities).

- There is no match between questions/hypotheses and data analyses. It is just a compilation of statistics and methods without a clear direction or purpose. Overall the data visualization and statistical reporting feels intentionally overly complicated. No 'fancy' data analysis can compensate for lack of clear questions/hypotheses.

- The authors do not seem to grasp the nature and implications of their relative abundance data.

- There is almost no real discussion, just repetition of the same results over and over.

- Some recommendations for data visualization:

1) Voronoi diagrams in Fig. 1 are an overkill and they hide uncertainty and number of samples. Barplot/boxplots/means with error bars would be easier to understand and would depict uncertainty.

2) In Fig. 1 map it would be great to see colors for croplands vs grasslands. That would help the reader understand the number of samples in one environment vs the other, and their geographical distribution (from the text sometimes it seems a paired design)

3) Fig. S2A seems the most relevant one. I would suggest moving it to the main text (and include the data points to assess number of samples)

4) Fig. 2A is confusing. I would recommend to do a common diversity plot, either a rarefaction curve, x axis should be number of individuals (or alternatively number of samples) and y axis the number of OTUs, or a rank-abundance, x axis should be rank abundance from the most abundant to the least abundant and y axis the relative abundance (usually log transformed). In Fig. 2B, commonness is generally referred as occupancy

MINOR COMMENTS

I provide some minor comments but at this stage, the manuscript should focus on the major structural issues described above.

L60: If the authors want to introduce agricultural intensification (or in the discussion section) perhaps it deserves a more nuanced discussion. Large area is not an intensive practice per se, the

benefits or problems depend on yield and also if measured by area or product unit, see:
<https://www.sciencedirect.com/science/article/pii/S0301479712004264>
<https://besjournals.onlinelibrary.wiley.com/doi/full/10.1111/1365-2664.12035>
Lower yields->higher food prizes-> more land brought into agricultural production
Anyway, some nuance would enrich the introduction and the discussion sections, and would overall improve the manuscript.

L74: In 2023 and with the amount of papers (including dozens of reviews) about microbial biogeography and community structure, initiating with Baas-Becking feels somehow naive.

L88: It is certainly true that there are more studies on plants and animals but some worth-mentioning efforts have been devoted to microbes. Just a couple of examples:
<https://www.sciencedirect.com/science/article/pii/S2351989421000718>
<https://online.ucpress.edu/elementa/article/8/1/005/112322/Associations-between-human-impacts-and-forest-soil>

L91: Biotic homogenization can occur by reducing the presence and abundance of rare taxa but also just by similar community composition across samples, without any links to alpha-diversity.

L113: p-value by itself means close to nothing. It is necessary to explicit test/model, statistic and sample size.

L116: Obviously. Refer to Figure S3

L117: Include stats. I assume is PERMANOVA, then include R2

L119: Where are the plots and the statistics to support this statement? This can be done with a nested PERMANOVA

L124: Without any reference to a figure or statistic, it is even hard to understand what is the cumulative abundance of functional groups

L131: For grasslands and croplands, it is basically the same

L136: It is not interesting at all, it is just a product of relative abundance data. When one dominant group goes up, the other dominant group has to go down. This is so obvious that makes me worry

L141: Missing the word 'grasslands'

L148: composition not distribution

L153: For grasslands r is reported, for croplands just the p-value

L159: I assume these p-values correspond to the chi-squared statistic

L161: The variables included explain more in grasslands, inferring ecological processes from R2 (explanatory power) might be too much of a stretch

L170: So confusing using the abundant 26 OTUs, just do a rarefaction curve, or show a Shannon diversity comparison. Too much text to describe an obvious (and expected) diversity pattern

L177: a resemblance? Do you mean a correlation?

L181: How can you possibly infer dispersal limitation from different composition?

L185-189: So confusing, I don't even know what is the purpose of these analyses or how they are different from what has been reported in the previous 20 lines

L192: Obviously if they are rare, they are specialists by your definition of specialists!

L204: Are these variables collinear?

L207: Significantly but weakly!

L212: Relative abundance data!

L215: This is key piece of information (156 croplands, 61 grasslands)! This should be in the Introduction and also in the Methods section

L217: You have not measured functional diversity, just say diversity.

L218: Both statements are fundamentally the same, higher diversity implies more rare species

L225: Keep conflating the same thing different ways

L229: Afterwards? There is no time axis in this study

L232: So far you are comparing croplands vs grasslands, not agricultural intensification per se

L250: Niche conservatism because two different environments are different?

L271: Keep p-values in the results section

L297: Obviously because you are deriving your functional groups from your taxonomic annotation!

L302: Fungal biography? 16 authors and nobody noticed?

L356: It would be interesting to see if there are differences in diversity and composition depending on crop (although it would be a highly unbalanced design)

L359: This description seems to indicate that it is a paired design

L415: This filtering step might be heavily influencing the results on rare taxa. I would strongly suggest to repeat analyses without this filtering step and assess if the main results hold

L422: Not sure why it is necessary to mention R Markdown
L427: This was already mentioned in L419
L430: What type of filtering was done for the funguild results?
L431: I would move this sentence to the figure caption
L448: Why?? The whole point of PCA is for variables to be correlated!
L452: Why go from PERMANOVA to CAP? They provide very similar answers
L456: Mantel tests description fits better in the previous paragraph
L463: But the authors have not used resources, you used sites to calculate niche breadth!
L473: The best model should be preferentially selected based on theory, not predictive fit. The authors want to make inferences, not optimize prediction. I would recommend to read this recent review:
<https://esajournals.onlinelibrary.wiley.com/doi/full/10.1002/ecy.3336>

Reviewer #2:

Remarks to the Author:

This is a potentially excellent manuscript documenting the effects of agricultural management intensity on soil fungal biodiversity. Whilst this is a topic that has been covered previously in several papers, there is novelty here in the scale of sampling (across Europe) and the molecular approach used (Pacbio amplicon sequencing). In terms of findings the main result is that arable lands appear to have less fungal diversity than grasslands, with the scale of the sampling giving this paper significant weight with respect to the broader evidence base pertinent for this particular question. Also the MS identifies rare taxa as being particularly sensitive to intensification, which adds to the novelty of the work. The paper is excellently written, and the data thoroughly and appropriately analysed.

Specific Comments:

General point – little information is given in the main text regarding sampling design. For all I know as a reader – all the arable sites could have been in Spain, and all the grasslands in Sweden. I appreciate from looking at the supplementary data that this is not the case, but more needs to be made of this in the main text I feel as this detail is critical for interpretation of the findings. Line 358 (M&M) elaborates somewhat by stating that the contrasts were neighbouring (great!), but then why is the number of arable (n=156) v grass (n=61) sites not equal? Given the often confounding effects of soil properties on land use effects, and the small effect size and error associated with the claims of less diversity in arable – assuring the reader that there are not confounding effects is paramount here. The extent and design of a “survey” type experiment is often critical in influencing the interpretation and claims that can be made from the results, and for a potentially high impact paper such as this it needs to be solid and transparent I feel.

Line 207 A line is slipped in here implying that rare taxa are positively associated with gross ecosystem functions. Intuitively this doesn't really make much sense, especially in soils where rare taxa may be dormant or even dead (extracellular DNA). So to me, a spurious association – grasslands happen to have more fungal rareness, and greater ecosystem functions (of those measured). If the authors are inferring that these elevated functions may be the result of more rareness...some sort of combined effect of rare taxa then this would indeed be a highly unexpected finding. However given the lack of causal evidence for this ...I'd be inclined to remove or downplay.

Reviewer #3:

Remarks to the Author:

This is a wonderful description of fungal communities in a transect across Europe. Who would not like some solid fungal analyses? Hence the results are for sure noteworthy! However, the presentation of the results and the discrepancy between abstract – results shown – and discussion is still (too) big. The results are solid but authors base their main interest (abstract/discussion) that are derived from many unclear steps ('fishing expedition') in results and essentially are not shown clearly in figures. The major aspects – that still the location determined the community

structure – is largely ignored and focus is on rare species. For this, a very good explanation on data handling and how the rare species (let alone their functionality!!) are determined. These are aspects that are mentioned as main results but one needs to dig really deep into supplementary materials to find answers. Also, some of the methods such as rarefaction need to be better justified as this really affects the conclusions on rare microbes – while for other questions (community of microbes) asked it is fairly standard method. Further, it would be good to look into this paper (albeit about bacteria):

Jia, X., Dini-Andreote, F. and Salles, J.F., 2022. Unravelling the interplay of ecological processes structuring the bacterial rare biosphere. *ISME Communications* 2, 1.

Overall, the manuscript has many good aspects but the message authors want to convey does not come across from the results and figures which leads to confusion when reading. Also, could be that the message is not there – at least in as strong manner as the authors would wish for.

More specific, but still major comments:

In abstract on line 49 it is said 'dominant groups become more dominant' but is this not related to the fact that abundances are relative and the data has been rarified.

Line 52- this is interesting result but not explained well in results section and the calculation for intensity not explained almost at all – which would be needed for one of the main results. Also, does this only apply to arable land or also to grasslands? Are they included in the model?

While I agree with the call to protect rare soil fungi, I am not fully convinced that the data here shows that they are responsible for main functions of the ecosystem and even if they will benefit from reducing intensity of agricultural practices. The effect of these rare fungi on ecosystem functions is not really tested and it is unknown if any rare fungi will be good or just some parts of the community.

When referring to arbuscular mycorrhizal fungi (or ericoid mycorrhizal fungi etc), please use the word fungi as mycorrhizae is the association. Mycorrhizal fungi are also shown consistently to be less important and abundant in agroecosystems (compared to more natural ecosystems) so it is not maybe such an important example.

The background information and theory on rarity is interesting and well presented. Indeed, rarity is important for some of the bacterial processes – is this why it would be expected to matter for fungi as well? Isn't this a bit contrary to the global fungal literature?

Resource heterogeneity will be one of the major differences in the system – it is quite a pity this is not measured and the degree of plant diversity within rotations and in case of grasslands, plots, is not included. This could be one of the major explanations. Plant species identity has been shown as one of the major factors affecting soil fungi – so it could be assumed that some plants would have unique members in their microbiome. If only very narrow range of plants (essentially wheat) was sampled for agricultural sites, and compared to diverse plots in grassland, this is expected outcome. Furthermore, in less intensive agricultural systems a broader crop rotation (or even strip cropping) is in use, it could be that plant diversity is the driver in larger scale.

Further, the design is quite unbalanced with more arable sites, how was this handled in the models? - Now it is 156 vs 61 and there were even more wheat plots than grasslands. This also affects the rarity calculations and the heterogeneity. Also, the plant diversity might be very different in marginal land than in native grassland.

Lines 117-119: I would say based on the figures that the site is THE important determinant of community structure.

The fungal classes detected are still relatively more or less abundant. It would be also nice to have statistics on this to see if it is consistent pattern. It is important to also note that the 'functional groups' are predicted functional groups based on sequencing data. For the functional groups, this also depends on the groups looked at. Many mycorrhizal groups will be absent because of lack of suitable hosts: this is hardly surprising when comparing ecosystems with each other.

Lines 130-143: long bit about something that has not been introduced in the introduction/hypotheses (or abstract). These seem to be also supplementary figures so please consider if this is really something the authors want to say. This is the point of abstract and results not being in line and the main message comes very late (if at all) in the manuscript. Overall this paragraph is very long and only the (partial) Mantel tests showing the relationship between soil chemistry, distances between sites and climatic factors and fungal community are meaningful for the story. This also hints towards the fact that authors have undersampled the system to draw some of the conclusions they have drawn. Particularly, if grasslands are more dissimilar in all properties, it does make sense that they have more dissimilar fungal communities and hence also more rare species of fungi. Then question remains is if this is because grasslands are not sampled

to enough depth or because they are more heterogenous environment.

Lines 165-169: It would be important to specify how rare taxa are determined, how data was handled and filtered here.

Line 182-184: it is not clear what this means. The occurrence of 'functional' groups is not maybe the main message here, is it? What is the conclusion on this topic, that there are some plant pathogens in all sites but ericoid mycorrhizal fungi are missing from arable soils due to lack of plants forming this association?

Line 193-195: But these are also very common taxa. So some species (whose function we do not know) belonging to these (normally very abundant) taxa are rare. I might ask why do we care? Either provide more details on species, or omit this.

195-195: Group shown to respond to plant diversity and fertilization

Line 205-206 is the main interesting result but now hidden inside the last bit of the results. The use of the index (and earlier also the rarity) should be also explained better.

Lines 207-208: This is again one of the main results, although the figure (S12) is not very convincing but rather looks like few outliers drive the pattern. What happens when 0s (how is the PcoA 0 for so many samples) are removed – or zero inflated models used. Spearman rank correlation might not be the best model for data with a lot of 0 close to each other.

Lines 209-212: Again interesting but not shown in figures and only mentioned briefly in the end of results.

Like stated earlier, the whole discussion should be aligned better with results (and vice versa) and could also benefit from shortening and focus. Especially statements like 'We advocate alleviation of intensive agricultural practices' is not really founded here to the data shown – and specific measures what to avoid might make sense.

Lines 408-409: If both ITS regions were sequenced, why short sequences (100bp) are included in the analyses? Would it not make sense to include only reads where both ITS regions are present (which would make them over 500bp?).

Line 413: How was ITSx used to verify taxonomic prediction – it is used usually to extract ITS regions from data.

Line 419: Was the rarity analyses also done on the rarified data?

Line 430: What kind of prediction probability was used? Was the data curated at all? What about multiple assignments?

Line 479-480: For something related to the main conclusion this is well hidden here.

The figures in general (besides fig 1) are unclear and not coherent with the message. Term functional rarity is odd. Further, not all functional groups are equally important (Fig 2C-D; pathogens of pollen, orchid mycorrhizae, undefined symbiotroph, animal endosymbiont) in both ecosystems for their functioning – and many are due to presence of herbivores and/or certain plants in grasslands while they are kept away from the arable fields in purpose. Figure 3 is difficult to understand, same for 4-B.

REVIEWER COMMENTS

Reviewer #1 (Remarks to the Author):

The research article 'Agricultural intensification is linked to the rarity of soil fungi across Europe' by Banerjee et al. describes soil fungal community patterns (mainly diversity) along a 3,000 km gradient. The authors compared fungal community diversity and composition between croplands and grasslands, and observed that grasslands are more diverse. It is my opinion that descriptive articles (particularly of large datasets) deserve to be published, when done right. This article suffers from overlapping and confusing analyses, poor data visualization, extremely deficient statistical reporting, and overinterpretation of results. Honestly, I doubt that all co-authors (leaders in their respective fields) have read and approved to submit this manuscript as it is. I am very sorry to say that the manuscript feels more like a first rough draft with a haphazard collection of data analyses, rather than a well-thought and revised version.

We have made numerous changes to the manuscript based on the comments from you and other reviewers. In particular, we have now changed the title, made analyses more coherent, improved data visualization, added extensive details on statistical analyses, toned down interpretations, and added clarifications. Additional figures and tables have been added to explain points. We have also added a section on the implications of our results and potential drawbacks.

Please find our point-by-point responses below.

To avoid confusion, all line numbers are based on the Revised Manuscript CLEAN version.

GENERAL COMMENTS

- Let's start with the title and their main result. The main result is about comparing croplands vs grasslands, and it is completely fine. Agricultural intensification only appears in the random forest analyses. You need to assess the collinearity between those variables. In my opinion, it would make much more sense if SEM analyses were used with the subset of cropland samples to answer the question about agricultural intensification (how climate and agricultural intensification both affect soil, and the direct and indirect effects to soil microbial communities).

Thank you for your suggestions. We have changed the title to: *"Lower soil fungal diversity in arable lands across Europe due to biotic homogenization and fewer rare taxa"*.

In the revised manuscript, we focus on land-use (arable vs grassland) and fungal biogeography, and how rare fungi are disproportionately affected in arable lands (Figures 3 and 4). We also show the geographic, edaphic, and climatic drivers of fungal biogeography (Figure 2, S10-S13). We used commonness and site occupancy to identify prevalent and rare fungi. The agricultural intensity index had a significant correlation with rare fungal diversity (Figure 3D). However, the correlation was weak and not adequate for a specific structural equation model targeting rare fungi. For the land-use SEM (Figure S13), we checked for collinearity and the variables were not influenced.

We identified the functional guilds that the rare taxa were associated with using FunGuild and FungalTraits databases (Figure 4B), and annotated their molecular functions with ITS sequences using the FunFun database (Figure S16). We found that the rare fungal groups were narrowly distributed, and the rarity was not proportional to the overall diversity. Finally, we show

that the diversity of rare fungi was positively and significantly correlated with important soil ecosystem functions, including soil aggregation, potential ammonification, and basal respiration (Figure S17), suggesting that the loss of such groups due to agricultural intensity might have functional implications. With this flow of analyses, we not only discuss the continental scale biogeography of soil mycobiome, but we also highlight the land use effect on rare fungal groups.

- There is no match between questions/hypotheses and data analyses. It is just a compilation of statistics and methods without a clear direction or purpose. Overall the data visualization and statistical reporting feels intentionally overly complicated. No 'fancy' data analysis can compensate for lack of clear questions/hypotheses. – The authors do not seem to grasp the nature and implications of their relative abundance data.

We have now explicitly stated our questions and hypotheses. Ln 101: *“Here, we aimed to address the following research questions: a) How do fungal diversity and community composition vary across land use and countries? b) What factors drive fungal biogeographical patterns at continental scale? and c) Are dominant and rare fungal groups differentially affected by land use?”*

Ln 114: *“We hypothesized that the effect of land use differs between dominant and rare fungal groups. Based on our findings from the root mycobiome²⁷, we also hypothesized that agricultural intensity has a negative influence on the overall soil fungal diversity. ”*

We have also rewritten the Discussion in terms of our hypotheses (Ln 181, Ln 186).

- There is almost no real discussion, just repetition of the same results over and over.

Following the suggestion from you and Reviewer 3, we have made numerous changes to the discussion and added new explanations (see for example, Ln 244-265; 296-337). We believe the discussion section is more succinct and in line with our results. We have also revised the conclusion section.

- Some recommendations for data visualization:

1) Voronoi diagrams in Fig. 1 are an overkill and they hide uncertainty and number of samples. Barplot/boxplots/means with error bars would be easier to understand and would depict uncertainty. 2) In Fig. 1 map it would be great to see colors for croplands vs grasslands. That would help the reader understand the number of samples in one environment vs the other, and their geographical distribution (from the text sometimes it seems a paired design)

Following the suggestion by the reviewer, we have now added a stacked barplot in the Supplementary Information, showing fungal community composition across two land-use types and five countries (Figure S3). Previous Figure 1 has been moved to the Supplementary Information to show the map of field sites.

3) Fig. S2A seems the most relevant one. I would suggest moving it to the main text (and include the data points to assess number of samples).

Thank you. We have added Fig S2 as the new Figure 1 in the main text.

4) Fig. 2A is confusing. I would recommend to do a common diversity plot, either a rarefaction curve, x axis should be number of individuals (or alternatively number of samples) and y axis

the number of OTUs, or a rank-abundance, x axis should be rank abundance from the most abundant to the least abundant and y axis the relative abundance (usually log transformed).

A rarefaction curve is already included in the manuscript (Fig S2).

We have now explained the cumulative abundance in detail further to avoid confusion.

Ln 165: “Grasslands had a proportionally higher presence of rare taxa compared to arable lands (**Figure 3**). Cumulative abundance was calculated by square-rooting the number of reads per sample, and then averaging and summing up the ordered relative abundances. The cumulative abundance was similar for the most abundant OTUs, but the difference increased for the rare (lower ranked) OTUs, with the square-rooting of the total number of sequences per site reaching 707.4 for grasslands compared with 589.4 for arable lands (**Figure 3A**). Arable lands had 16.7% fewer of fungal OTUs than grasslands and this was largely due to fewer rare OTUs. Furthermore, 75% of the total reads in arable lands was made up of the 19 most prevalent OTUs, while it required the 30 most prevalent OTUs to achieve 75% reads in grasslands. The cumulative abundance slopes for the two land use types were similar near the origin due to the prevalent OTUs, but they diverged due to fewer less abundant OTUs in the arable lands (**Figure 3A**), and this pattern was consistent across all countries (**Figure S14**). There was a positive correlation between OTU abundance and spatial commonness across sites (the number of sites the OTUs were present at) with rare OTUs significantly (t -test $p < 0.05$) more common in grasslands (**Figure 3B**). Agricultural intensity, calculated²⁸ based on the tillage intensity and the application of chemical fertilizers pesticides, was negatively associated with both the overall fungal richness ($\rho = -0.27$; $P < 0.001$) as well as the richness of rare taxa ($\rho = -0.24$; $P < 0.004$). Consistent with our hypothesis, these results suggest a possible negative impact of agricultural intensification on the soil mycobiome (**Figure 3 C & D**). On average, 40.7% of all OTUs were present in both land use types, whereas 19.5% and 39.8% were specific to arable lands and grasslands, respectively (**Supplementary Table 1**).”

In Fig. 2B, commonness is generally referred as occupancy

We have now mentioned this in the revised manuscript (Ln 178) and Figure 3.

MINOR COMMENTS

L60: If the authors want to introduce agricultural intensification (or in the discussion section) perhaps it deserves a more nuanced discussion. Large area is not an intensive practice per se, the benefits or problems depend on yield and also if measured by area or product unit, see:

<https://www.sciencedirect.com/science/article/pii/S0301479712004264>

<https://besjournals.onlinelibrary.wiley.com/doi/full/10.1111/1365-2664.12035>

Lower yields->higher food prizes-> more land brought into agricultural production. Anyway, some nuance would enrich the introduction and the discussion sections, and would overall improve the manuscript.

Thank you. We have used these papers to revise this part.

Ln 60: “However, there is often a trade-off between agricultural production and biodiversity, i.e., intensive agriculture can disrupt belowground communities and result in biodiversity loss^{5,6}. Studies performed at local scales have found a negative impact of intensive farming on the richness of specific fungal groups such as arbuscular mycorrhizal fungi^{4,7,8}. However, the effect of arable farming on the overall soil fungi should also be determined at large scales. Owing to the difference in the levels of resources and inputs in ecosystems (e.g., arable lands vs.

grasslands), the response of different functional groups to land-use change might also vary. The impact of production systems on biodiversity is not dichotomous, and land use effects on ecological communities can be nuanced and require large-scale investigations^{9,10}. Assessing such effects of land-use intensification on soil fungal biogeography across a wide range of climatic and soil conditions, and understanding its consequences for agroecosystem functioning is important. Such information can be used for designing sustainable farming practices that support fungi-mediated ecosystem services and bolster food security.”

L74: Naïven 2023 and with the amount of papers (including dozens of reviews) about microbial biogeography and community structure, initiating with Baas-Becking feels somehow naive.

We have deleted that sentence.

L88: It is certainly true that there are more studies on plants and animals but some worth-mentioning efforts have been devoted to microbes. Just a couple of examples:

<https://www.sciencedirect.com/science/article/pii/S2351989421000718>

<https://online.ucpress.edu/elementa/article/8/1/005/112322/Associations-between-human-impacts-and-forest-soil>

We have included these papers for our discussion on biogeography.

Ln 73: “Consequently, there has been a surge of fungal biogeography studies at large scales^{4,8,11–14}, revealing groups of fungi that are rare or restricted to a few sites. Beyond investigating the overall distribution, a fundamental goal of microbial biogeography studies is to identify the environmental factors that shape microbial distribution patterns^{15–18}. Identifying such factors can also reveal why some fungal groups display environmental filtering or provincialism while others are more widespread^{1,19}, helping develop models on soil fungal responses to land-use intensification at large scales.”

L91: Biotic homogenization can occur by reducing the presence and abundance of rare taxa but also just by similar community composition across samples, without any links to alpha-diversity.

The reviewer is right that biotic homogenization can be observed due to similar community composition across samples without an overall impact on alpha diversity. However, in our study, we did find an effect at the alpha diversity level i.e., all three indices tested employed here were lower in arable lands than grasslands (Ln 123, Figure 1).

Nonetheless, we have revised that sentence (Ln 93): *“Intensive management practices may exert a strong homogenizing effect on fungal communities, reducing the occurrence of rare taxa^{5,25}. Such biotic homogenization would result in a lower fungal diversity in intensively managed ecosystems^{6,14} with implications for fungi-mediated processes.”*

L113: p-value by itself means close to nothing. It is necessary to explicit test/model, statistic and sample size.

Added now (Ln 121)

L116: Obviously. Refer to Figure S3

Referred now (Ln 125).

L117: Include stats. I assume is PERMANOVA, then include R2

We included all values in the revised manuscript (Ln 130).

L119: Where are the plots and the statistics to support this statement? This can be done with a nested PERMANOVA

We indeed conducted a nested PERMANOVA, however, the results were not included in the previous version for brevity. We have now included all values in the revised manuscript. Ordination plots also included as **Supp Figure 5**.

The revised section reads (Ln 126): *“Geographic proximity exerted a significant effect (PERMANOVA $F = 14.369$; $R^2 = 0.206$; $P < 0.001$) on fungal communities with sites clustering based on their distribution from Sweden in the north to Spain in the south (Figure 1B, S5, S6). Within each cluster, the arable lands and grasslands also tended to group significantly (PERMANOVA $F = 11.369$; $R^2 = 0.040$; $P < 0.001$). The interactive effect of country and land use was also significant (PERMANOVA $F = 2.335$; $R^2 = 0.032$; $P < 0.001$).”*

L124: Without any reference to a figure or statistic, it is even hard to understand what is the cumulative abundance of functional groups

We have revised that sentence (Ln 135) to: *“Several functional groups of soil fungi also varied significantly across the five European countries (Figure 1D).”*

L131: For grasslands and croplands, it is basically the same

We have deleted that sub-sentence.

L136: It is not interesting at all, it is just a product of relative abundance data. When one dominant group goes up, the other dominant group has to go down. This is so obvious that makes me worry

We have deleted this sentence.

L141: Missing the word 'grasslands'

Corrected

L148: composition not distribution

Corrected

L153: For grasslands r is reported, for croplands just the p-value

We have included it for arable as well- ($r_{\text{Clim|Geo}} = 0.03$, $P = 0.23$)

L159: I assume these p-values correspond to the chi-squared statistic

Yes, we have clarified it now (Ln 157).

L161: The variables included explain more in grasslands, inferring ecological processes from R2 (explanatory power) might be too much of a stretch

We have deleted that sub-sentence.

L170: So confusing using the abundant 26 OTUs, just do a rarefaction curve, or show a Shannon diversity comparison. Too much text to describe an obvious (and expected) diversity pattern

We have deleted that sentence as suggested. Rarefaction curves (Figure S1) and Shannon diversity (Figure S2) have now been included.

L177: a resemblance? Do you mean a correlation?

We have revised that sentence to clarify the pattern (Ln 176): *“There was a positive correlation between OTU abundance and site occupancy or commonness (the number of sites the OTUs were present at) with rare OTUs significantly (t -test $p < 0.05$) more common in grasslands (Figure 3B).”*

L181: How can you possibly infer dispersal limitation from different composition?

We have deleted that sentence.

L185-189: So confusing, I don't even know what is the purpose of these analyses or how they are different from what has been reported in the previous 20 lines

We have deleted these lines and moved the mycorrhiza sentence up.

L192: Obviously if they are rare, they are specialists by your definition of specialists!

The revised manuscript does not have this sentence.

L204: Are these variables collinear?

No, they are not.

L207: Significantly but weakly!

We have now revised it (Ln 203): *“Rare soil fungi were also positively and significantly correlated with important soil ecosystem functions across both land-use types (Figure S18), indicating that the loss of such groups due to agricultural intensity might have functional implications. However, since these groups were a part of the rare mycobiome, the correlations were weak.”*

We have also moved that figure to the Supplementary Information.

L212: Relative abundance data!

We have revised the sentence to highlight this (Ln 208): “Overall, our results show a similarity between relative abundance and commonness, with dominant groups being more prevalent, while rare groups even rarer.”

L215: This is key piece of information (156 croplands, 61 grasslands)! This should be in the Introduction and also in the Methods section

We have added this in the Introduction (Ln 107), Discussion (Ln 213), and Methods (Ln 362).

L217: You have not measured functional diversity, just say diversity.

Corrected it (Ln 215).

L218: Both statements are fundamentally the same, higher diversity implies more rare species

Revised to (Ln 215): “We show that fungal diversity is consistently lower in arable lands than in grasslands across five countries. Rare fungi were affected or absent in arable lands, suggesting that biotic homogenization and a disproportionately negative impact of arable farming on the rare soil microorganisms”

L225: Keep conflating the same thing different ways

We have now deleted this sentence.

L229: Afterwards? There is no time axis in this study

We have revised this sentence as (Ln 173): “The cumulative abundance slopes for the two land use types were similar near the origin due to the prevalent OTUs, but they diverged due to fewer less abundant OTUs in the arable lands (Figure 3A), and this pattern was consistent across all countries (Figure S14).”

L232: So far you are comparing croplands vs grasslands, not agricultural intensification per se

We have shown the negative influence of agricultural management intensity on soil fungal diversity (Figure 3C) and discussed in the text.

Ln 220: “We found that, on an average, arable lands across the five European countries had nearly 25% lower fungal diversity than grasslands. One of the major impacts of intensive agriculture is the loss of biodiversity^{5,6}. Practices such as excessive use of synthetic fertilizers and pesticides, monocultures, excessive tillage, and the homogenization of landscapes can negatively affect the local and regional pool of biodiversity^{5,29}. Indeed, agricultural intensification has been linked to a systematic decline in birds, invertebrates and amphibians in arable lands^{6,30}. Importantly, biodiversity loss is not just a decrease in species number, but it also accompanies the loss of associations among various species, with the potential disruption of the network of mutual dependencies between species. For example, our previous report found that agricultural intensification has a negative impact on root endophytic fungi, and the associations among fungal members in conventional farmlands is 50% less than in organic lands²⁷. Recent studies also found that the abundance of beneficial mycorrhizal fungi is negatively associated with pesticide residues³¹ and pesticide application reduces the richness of mycorrhizal fungi and their ability to acquire phosphorus from the soil³². Despite this, we have limited knowledge of how intensive agricultural practices affect soil fungal diversity and distribution at large

scales³³. Here, we show that fungal diversity peaked at mid-latitude, although it was consistently lower in arable lands. There was a negative relationship between fungal diversity and the agricultural management intensity calculated from tillage, agrochemicals, and pesticide information (**Figure 3C and S16**). This is in contrast to a recent study which found that fungal diversity was similar in grasslands and non-permanent croplands¹⁴, which could be due to the wider range of crops (over 20 different crops including cereals and vegetables) sampled in that study. Most of the arable lands assessed in this study practiced conventional tillage, which has been found to negatively affect mycelial networks³⁴, and may be a cause of the lower fungal diversity in arable soils. Furthermore, fertilization applications in arable lands can cause resource homogeneity, which may result in the dominance of copiotrophic microorganisms. Arable systems also have fewer hosts for symbiotic groups. Extensively managed grasslands are an important type of agricultural system that covers nearly 16% of all lands in Europe³⁵. Farmers are encouraged and often required to maintain a certain proportion of their lands as grasslands in order to receive government subsidies³⁶. The consistently higher fungal diversity in grasslands supports the maintenance of grasslands as part of the farmlands to promote the soil mycobiome. We also found a greater abundance of mycorrhizal fungi and a relatively lower abundance of pathogens in grasslands. Overall, our results show that soil mycobiome displays contrasting biogeographical patterns between land-use types but a consistency across the 3000-km European gradient.”

L250: Niche conservatism because two different environments are different?

We have now deleted this sub-sentence.

L271: Keep p-values in the results section

Done

L297: Obviously because you are deriving your functional groups from your taxonomic annotation!

The revised manuscript does not have this sentence.

L302: Fungal biogeography? 16 authors and nobody noticed?

We have corrected it to 'biogeography' (Ln 270)

L356: It would be interesting to see if there are differences in diversity and composition depending on crop (although it would be a highly unbalanced design)

Yes, we did explore this, but due to the unbalanced design we have avoided to refer to this in the discussion.

L359: This description seems to indicate that it is a paired design

The revised sentence reads (Ln 366): “When possible, we paired agricultural fields with non-arable lands by sampling nearby extensively managed grasslands and marginal lands with permanent, predominantly herbaceous plant cover. These non-arable sites were mostly unfertilized and occasionally mowed.”

L415: This filtering step might be heavily influencing the results on rare taxa. I would strongly suggest to repeat analyses without this filtering step and assess if the main results hold

Sorry for the confusion. We actually did not do this filtering and only removed the singletons. We will upload the entire R script to show this.

L422: Not sure why it is necessary to mention R Markdown

Removed (Ln 435)

L427: This was already mentioned in L419

We would like to state the R package used, so we have revised both sentences (Ln 430 and 438).

L430: What type of filtering was done for the funguild results?

We only removed the singletons and unidentified taxa.

L431: I would move this sentence to the figure caption

Moved

L448: Why?? The whole point of PCA is for variables to be correlated!

We did this because strongly correlated variables will produce similar results anyway.

L452: Why go from PERMANOVA to CAP? They provide very similar answers

Since CAP is a constrained form of ordination, we constrained the fungal data only after ascertaining the significant effect through PERMANOVA. This is mentioned in Ln 461.

L456: Mantel tests description fits better in the previous paragraph

Done

L463: But the authors have not used resources, you used sites to calculate niche breadth!

We have revised this sentence to (Ln 483): *“Rare taxa were identified from the original OTU table by identifying OTUs that are only present at one or two sites of respective land use types and have a narrow niche breadth (<0.55). We computed niche breadth using the MicroNiche package⁵⁵, which calculates the proportional occurrence of taxa across sites.”*

L473: The best model should be preferentially selected based on theory, not predictive fit. The authors want to make inferences, not optimize prediction. I would recommend to read this recent review:

<https://esajournals.onlinelibrary.wiley.com/doi/full/10.1002/ecy.3336>

Indeed, we preferentially selected the model based on parsimony. Coefficient of determination, and goodness of fit. We have also read and cited this article.

Ln 467: *“We performed structural equation modelling (SEM) to investigate complex relationships among geographical locations, climatic conditions, soil properties and fungal communities using the lavaan package version 0.6-3⁵². An initial SEM was constructed based on the understanding of factors that shape soil fungal distribution (Supplementary Figure 13). Briefly, geographic locations influence soil fungal diversity through fungal dispersal limitations. Climate and soil properties are the two environmental filters on soil fungal communities. Texture, isothermality and mean temperature in the wettest season were also removed to achieve better model fit and parsimony. Samples containing missing values were removed, resulting in 143 observations for arable lands and 54 observations for grasslands, with 8 degrees of freedom. An initial SEM was formulated (Figure S13) and modified sequentially by removing links that were not significant and hindered model fit, or by adding links that had high modification indices. The final SE model had adequate fit (parable lands = 0.294; pgrasslands = 0.699; Figure 16). A model was considered acceptable when the chi-square test p-value was greater than 0.05. The best model was preferentially selected based on parsimony, coefficient of determination of fungal diversity and goodness of fit using the maximum likelihood approach⁵³.”*

Reviewer #2 (Remarks to the Author):

This is a potentially excellent manuscript documenting the effects of agricultural management intensity on soil fungal biodiversity. Whilst this is a topic that has been covered previously in several papers, there is novelty here in the scale of sampling (across Europe) and the molecular approach used (Pacbio amplicon sequencing). In terms of findings the main result is that arable lands appear to have less fungal diversity than grasslands, with the scale of the sampling giving this paper significant weight with respect to the broader evidence base pertinent for this particular question. Also the MS identifies rare taxa as being particularly sensitive to intensification, which adds to the novelty of the work. The paper is excellently written, and the data thoroughly and appropriately analysed.

Thank you for your positive assessment of our manuscript.

Specific Comments:

General point – little information is given in the main text regarding sampling design. For all I know as a reader – all the arable sites could have been in Spain, and all the grasslands in Sweden. I appreciate from looking at the supplementary data that this is not the case, but more needs to be made of this in the main text I feel as this detail is critical for interpretation of the findings.

We agree with the reviewer. We have now provided clarifications on the sampling design the Introduction (Ln 109), Discussion (Ln 214), and Methods (Ln 366). Site information is also shown in Figure S1.

Line 358 (M&M) elaborates somewhat by stating that the contrasts were neighbouring (great!), but then why is the number of arable (n=156) v grass (n=61) sites not equal? Given the often confounding effects of soil properties on land use effects, and the small effect size and error associated with the claims of less diversity in arable – assuring the reader that there are not confounding effects is paramount here. The extent and design of a “survey” type experiment is often critical in influencing the interpretation and claims that can be made from the results, and for a potentially high impact paper such as this it needs to be solid and transparent I feel.

First of all, as stated above, we have now explicitly stated the sample number difference in the revised manuscript to make it clear.

Secondly, we have now clarified this in the Methods (Ln 366): *“When possible, we paired agricultural fields with non-arable lands by sampling nearby extensively managed grasslands and marginal lands with permanent, predominantly herbaceous plant cover. These non-arable sites were mostly unfertilized and occasionally mowed. Many of these arable lands did not have nearby extensively managed grasslands, which resulted in an unbalanced design.”*

Finally, we did compute fungal richness on the same number of arable fields (n=61) by randomly selecting fields. Fungal richness was still lower for arable fields. Ln 370: *“However, to address whether the unbalanced design affected the outcome of this study, we computed fungal richness on the same number of arable lands (n=61) by randomly selecting fields. Our analysis revealed that fungal richness was still lower for arable lands than grasslands (Figure S4).”*

Line 207 A line is slipped in here implying that rare taxa are positively associated with gross ecosystem functions. Intuitively this doesn't really make much sense, especially in soils where rare taxa may be dormant or even dead (extracellular DNA). So to me, a spurious association – grasslands happen to have more fungal rareness, and greater ecosystem functions (of those measured). If the authors are inferring that these elevated functions may be the result of more rareness...some sort of combined effect of rare taxa then this would indeed be a highly unexpected finding. However given the lack of causal evidence for this ...I'd be inclined to remove or downplay.

The correlations were across both land-use types, and not just for grasslands. We have rephrased that sentence to (Ln 203): *“Rare soil fungi were also positively and significantly correlated with important soil ecosystem functions across both land-use types (Figure S18), indicating that the loss of such groups due to agricultural intensity might have functional implications. However, since these groups were a part of the rare mycobiome, the correlations were weak.”*

We have also moved that figure to the Supplementary Information.

We also agree with the reviewer that rare taxa can be dormant microbial members or part of the relic DNA pool in soil. We have now added this clarification in the Discussion (Ln 326). *“It is important to note that rare taxa can be dormant microbial members or a part of the relic DNA pool in the soil. Future studies may wish to identify active rare taxa by treating DNA samples with propidium monoazide (PMA), which binds to relic DNA and prevents subsequent amplification⁴⁵.”*

Reviewer #3 (Remarks to the Author):

This is a wonderful description of fungal communities in a transect across Europe. Who would not like some solid fungal analyses? Hence the results are for sure noteworthy! However, the presentation of the results and the discrepancy between abstract – results shown – and discussion is still (too) big. The results are solid but authors base their main interest (abstract/discussion) that are derived from many unclear steps ('fishing expedition') in results and essentially are not shown clearly in figures.

We thank the reviewer for the positive assessment of our manuscript. We have now addressed your concerns in the revised manuscript. Please find our point-by-point responses below.

The major aspects – that still the location determined the community structure – is largely ignored and focus is on rare species. For this, a very good explanation on data handling and how the rare species (let alone their functionality!!) are determined. These are aspects that are mentioned as main results but one needs to dig really deep into supplementary materials to find answers. Also, some of the methods such as rarefaction need to be better justified as this really affects the conclusions on rare microbes – while for other questions (community of microbes) asked it is fairly standard method.

Further, it would be good to look into this paper (albeit about bacteria):

Jia, X., Dini-Andreote, F. and Salles, J.F., 2022. Unravelling the interplay of ecological processes structuring the bacterial rare biosphere. ISME Communications 2, 1.

Overall, the manuscript has many good aspects but the message authors want to convey does not come across from the results and figures which leads to confusion when reading. Also, could be that the message is not there – at least in as strong manner as the authors would wish for.

Thank you for your critical and constructive comments. In the revised manuscript, we focused on the influence of locations and land-use (arable vs grassland) types on fungal biogeography. This is now further addressed, also in the results section (Ln 122-162). We then focused on rare fungi and how they are disproportionately affected in arable lands (Figure 4; S15-18). We also showed the geographic, edaphic, and climatic drivers of overall fungal biogeography (Figure 2, S9-S13). We gave attention to location, which also had a strong effect on fungal community structure (Ln 165-184). However, even after correcting for location (country) effects and sample numbers (Figure S4), land-use type had an impact on fungal rarity with biotic homogenization in arable lands.

We identified the functional guilds that the rare taxa were associated with using FunGuild and FungalTraits databases (Figure 4B), and their molecular functions annotated with ITS sequences using the FunFun database (Figure S18). We have also explicitly stated our data handling, how rare fungi were identified, and how we annotated the functional groups (Ln 484).

Thank you for suggesting this useful article. We have used it for our discussion on rare taxa.

Additionally, we have added numerous clarifications and details to avoid confusion.

More specific, but still major comments:

In abstract on line 49 it is said 'dominant groups become more dominant' but is this not related to the fact that abundances are relative and the data has been rarified.

We agree with you regarding relative abundance derived from sequencing datasets. In the revised manuscript, we focused on prevalence and rarity. Throughout the manuscript, we adhere to commonness and site occupancy to identify prevalent and rare fungi. That sentence in the abstract now reads (Ln 50): “Prevalent fungal groups became even more abundant, whereas rare groups became fewer or absent in arable lands, suggesting biotic homogenization due to arable farming.”

Line 52- this is interesting result but not explained well in results section and the calculation for intensity not explained almost at all – which would be needed for one of the main results. Also, does this only apply to arable land or also to grasslands? Are they included in the model?

We have now provided additional details on the calculation of agricultural intensity index. It was only calculated for the arable sites.

Ln 495: “ *We created a single index of management intensity that could be compared across arable sites and countries, and this index was used in a previous study²⁸. For this, we used the management data obtained from the 2017 growing season. The specific management variables that were used in this index were the number of insecticide, herbicide and fungicide applications, the number of tillage events, the maximum tillage depth, and the total amount of mineral nitrogen applied. All of the above parameters were then included in a single management index using principal component analysis and scaling the index between 0 and 1, with 0 indicating the minimum intensity of management practices and 1 indicating the maximum intensity.*”

While I agree with the call to protect rare soil fungi, I am not fully convinced that the data here shows that they are responsible for main functions of the ecosystem and even if they will benefit from reducing intensity of agricultural practices. The effect of these rare fungi on ecosystem functions is not really tested and it is unknown if any rare fungi will be good or just some parts of the community.

We have toned down our discussion and moved that figure to the Supplementary Information.

When referring to arbuscular mycorrhizal fungi (or ericoid mycorrhizal fungi etc), please use the word fungi as mycorrhizae is the association. Mycorrhizal fungi are also shown consistently to be less important and abundant in agroecosystems (compared to more natural ecosystems) so it is not maybe such an important example.

Thank you. We used ‘mycorrhizal fungi’ throughout the revised manuscript.

The background information and theory on rarity is interesting and well presented. Indeed, rarity is important for some of the bacterial processes – is this why it would be expected to matter for fungi as well? Isn’t this a bit contrary to the global fungal literature?

While there is limited information on the importance of rare fungal taxa, we have provided examples to highlight their importance.

Resource heterogeneity will be one of the major differences in the system – it is quite a pity this is not measured and the degree of plant diversity within rotations and in case of grasslands, plots, is not included. This could be one of the major explanations. Plant species identity has been shown as one of the major factors affecting soil fungi – so it could be assumed that some

plants would have unique members in their microbiome. If only very narrow range of plants (essentially wheat) was sampled for agricultural sites, and compared to diverse plots in grassland, this is expected outcome. Furthermore, in less intensive agricultural systems a broader crop rotation (or even strip cropping) is in use, it could be that plant diversity is the driver in larger scale.

Thank you for your comment. It is true that plant diversity was not measured in this study, and it might have been the determinant of observed patterns. We have now explicitly stated this in the Discussion.

Ln 222: *“Practices such as excessive use of synthetic fertilizers and pesticides, monocultures, excessive tillage, and the homogenization of landscapes can negatively affect the local and regional pool of biodiversity^{5,29}. Indeed, agricultural intensification has been linked to a systematic decline in birds, invertebrates and amphibians in arable lands^{6,30}. Importantly, biodiversity loss is not just a decrease in species number, but it also accompanies the loss of associations among various species, with the potential disruption of the network of mutual dependencies between species. For example, our previous report found that agricultural intensification has a negative impact on root endophytic fungi, and the associations among fungal members in conventional farmlands is 50% less than in organic lands²⁷. Recent studies also found that the abundance of beneficial mycorrhizal fungi is negatively associated with pesticide residues³¹ and pesticide application reduces the richness of mycorrhizal fungi and their ability to acquire phosphorus from the soil³². Despite this, we have limited knowledge of how intensive agricultural practices affect soil fungal diversity and distribution at large scales³³. Here, we show that fungal diversity peaked at mid-latitude, although it was consistently lower in arable lands. There was a negative relationship between fungal diversity and the agricultural management intensity calculated from tillage, agrochemicals, and pesticide information (**Figure 3C and S16**). This is in contrast to a recent study which found that fungal diversity was similar in grasslands and non-permanent croplands¹⁴, which could be due to the wider range of crops (over 20 different crops including cereals and vegetables) sampled in that study. Most of the arable lands assessed in this study practiced conventional tillage, which has been found to negatively affect mycelial networks³⁴, and may be a cause of the lower fungal diversity in arable soils. Furthermore, fertilization applications in arable lands can cause resource homogeneity, which may result in the dominance of copiotrophic microorganisms. Arable systems also have fewer hosts for symbiotic groups. Extensively managed grasslands are an important type of agricultural system that covers nearly 16% of all lands in Europe³⁵. Farmers are encouraged and often required to maintain a certain proportion of their lands as grasslands in order to receive government subsidies³⁶. The consistently higher fungal diversity in grasslands supports the maintenance of grasslands as part of the farmlands to promote the soil mycobiome. We also found a greater abundance of mycorrhizal fungi and a relatively lower abundance of pathogens in grasslands. Overall, our results show that soil mycobiome displays contrasting biogeographical patterns between land-use types but a consistency across the 3000-km European gradient. However, plant community composition was not measured for grassland sites in this study and as a result, the importance of plant diversity could not be tested in our study. Moreover, we sampled extensive grasslands, which are different from pristine native grasslands in terms of their species composition. As shown in previous studies⁴¹, plant diversity can be an important determinant of soil fungi as a diversity plant community influences soil physical properties with different root architectures, shapes the soil chemistry with diverse root exudates and residues, and thereby modulates the soil mycobiome. For example, labile resources will attract copiotrophic fungi while oligotrophic groups will settle for more recalcitrant resources. Indeed, resource heterogeneity may also vary based on the crop rotations in arable*

lands⁴². Future studies should measure plant diversity when arable to native ecosystems to dissect above- and belowground linkages.”

Further, the design is quite unbalanced with more arable sites, how was this handled in the models?- Now it is 156 vs 61 and there were even more wheat plots than grasslands. This also affects the rarity calculations and the heterogeneity. Also, the plant diversity might be very different in marginal land than in native grassland.

Rarefaction curve is included in the manuscript (Fig S1). We have calculated fungal species richness (total number of species) for the same number of arable and grassland sites. Fungal richness was still significantly higher in grassland soils. Please see Figure S4

As mentioned above, we have now discussed the importance of plant diversity and resource heterogeneity in shaping the soil mycobiome.

Lines 117-119: I would say based on the figures that the site is THE important determinant of community structure.

We have revised that sentence (Ln 127) to: “Geographic proximity emerged as one of the most important factors and exerted a significant effect (PERMANOVA $F = 14.369$; $R^2 = 0.206$; $P < 0.001$) on fungal communities with sites clustering based on their distribution from Sweden in the north to Spain in the south (Figure 1B, S5, S6).”

The fungal classes detected are still relatively more or less abundant. It would be also nice to have statistics on this to see if it is consistent pattern. It is important to also note that the ‘functional groups’ are predicted functional groups based on sequencing data. For the functional groups, this also depends on the groups looked at. Many mycorrhizal groups will be absent because of lack of suitable hosts: this is hardly surprising when comparing ecosystems with each other.

We have added a stacked barplot with error bars showing fungal community composition across two land-use types and five countries (Figure S7). Furthermore, we have included two tables in the Supplementary Information (Table S2 and S3) showing the members of major fungal classes that changed significantly across two land-use types and five countries. Ln 134

In the revised manuscript, we identified the functional guilds that the rare taxa were associated with using FunGuild and FungalTraits databases (Figure 4B), and their molecular functions annotated with ITS sequences using the FunFun database (Figure S17). Finally, we show that the diversity of rare fungi was positively and significantly correlated with important soil ecosystem functions, including soil aggregation, potential ammonification, and basal respiration (Figure S18).

Lines 130-143: long bit about something that has not been introduced in the introduction/hypotheses (or abstract). These seem to be also supplementary figures so please consider if this is really something the authors want to say. This is the point of abstract and results not being in line and the main message comes very late (if at all) in the manuscript. Overall this paragraph is very long and only the (partial) Mantel tests showing the relationship between soil chemistry, distances between sites and climatic factors and fungal community are meaningful for the story. This also hints towards the fact that authors have undersampled the system to draw some of the conclusions they have drawn. Particularly, if grasslands are more dissimilar in all properties, it does make sense that they have more dissimilar fungal

communities and hence also more rare species of fungi. Then question remains is if this is because grasslands are not sampled to enough depth or because they are more heterogenous environment.

We appreciate your comment. Firstly, we have deleted several sentences form that section to make it more succinct. Secondly, our aim was to not only report continental scale patterns of soil fungal distribution but also to elucidate the drivers of biogeography. Thus, we have now added a figure (Figure 2) highlighting the edaphic, geographical, and climatic factors that shaped soil mycobiome across Europe. We have also discussed the importance of these factors in the revised manuscript (Ln 142-162; Ln 268-293). Finally, as mentioned above, we have now compared fungal diversity for the same number of samples and found similar values of diversity as the ones reported (Figure S4).

Lines 165-169: It would be important to specify how rare taxa are determined, how data was handled and filtered here.

We have explicitly stated our data handling, how rare fungi were identified, and how we annotated the functional groups (Ln 483).

Line 182-184: it is not clear what this means. The occurrence of 'functional' groups is not maybe the main message here, is it? What is the conclusion on this topic, that there are some plant pathogens in all sites but ericoid mycorrhizal fungi are missing from arable soils due to lack of plants forming this association?

We have now shortened the section to improve clarity. In particular, we have now deleted this sentence. The sentence on mycorrhiza reads (Ln 136): "*Mycorrhizal fungal groups (ecto, arbuscular, orchid, and Ericoid) showed higher relative abundance in grasslands than in arable lands, which may be due to the availability of host plants in grasslands.*"

Line 193-195: But these are also very common taxa. So some species (whose function we do not know) belonging to these (normally very abundant) taxa are rare. I might ask why do we care? Either provide more details on species, or omit this.

Most of these groups were unclassified at the genus and species level, but we have provided example of some general part of the rare pool. Secondly, we have provided a section on the importance of rare fungi and why their loss might have functional implications.

Ln 192: "*Several orders of the rare taxa were more common in grasslands than arable lands, including Pleosporales, Thelephorales, Pezizales, and Hypocreales, which are well-known for their response to plant diversity (Figure 4B). While a large majority of rare fungi were unclassified at the genus level, we found noticeable presence of member of Mortierella, Archaeorhizomyces, Entoloma, Pluteus, and Psathyrella genera.*"

195-195: Group shown to respond to plant diversity and fertilization

We have added this point (Ln 194). Thanks

Line 205-206 is the main interesting result but now hidden inside the last bit of the results. The use of the index (and earlier also the rarity) should be also explained better.

Thank you. We have now added it in Figure 3 and S16. and discussed the results in terms of our hypothesis. Please check Ln 181.

Lines 207-208: This is again one of the main results, although the figure (S12) is not very convincing but rather looks like few outliers drive the pattern. What happens when 0s (how is the PcoA 0 for so many samples) are removed – or zero inflated models used. Spearman rank correlation might not be the best model for data with a lot of 0 close to each other.

Those values near 0 are negative values. To reassess this, we added a constant and performed squared-root transformation. We then calculated Bray-Curtis dissimilarity and performed PCoA. However, the patterns still remained the same. This is now included as Figure S18.

Lines 209-212: Again interesting but not shown in figures and only mentioned briefly in the end of results.

The revised manuscript does not have this sentence.

Like stated earlier, the whole discussion should be aligned better with results (and vice versa) and could also benefit from shortening and focus. Especially statements like 'We advocate alleviation of intensive agricultural practices' is not really founded here to the data shown – and specific measures what to avoid might make sense.

We have deleted that sentence and made numerous changes to the Discussion to discuss the main results and implications of this study. Please check.

Lines 408-409: If both ITS regions were sequenced, why short sequences (100bp) are included in the analyses? Would it not make sense to include only reads where both ITS regions are present (which would make them over 500bp?).

We agree with the reviewer. The majority of the OTUs ranged 500 bp and 700 bpm and there was almost nothing under 400 bp. Please see below.

Line 413: How was ITSx used to verify taxonomic prediction – it is used usually to extract ITS regions from data.

Apologies for the confusion. Taxonomic assignments were derived from the UNITE database using the SINTAX algorithm. We have clarified that (Ln 425).

Line 419: Was the rarity analyses also done on the rarified data?

No, it was not. We used the original OTU table for that. Now clarified it in the text (Ln 483): *“Rare taxa were identified from the original OTU table by identifying OTUs that are only present in one or two sites at respective land use types.”*

Line 430: What kind of prediction probability was used? Was the data curated at all? What about multiple assignments?

Thank you. We used FungalTraits to corroborate the results obtained from FunGuild. In case of multiple assignments, we chose the first assigned group. The revised section reads (Ln 490): *“We used the FunGuild database⁶⁵ to identify the major functional groups that these rare taxa belong to and verified the functional guilds using the FungalTraits database⁷¹. To assess the cellular and molecular functions of rare fungi, we used FunFun⁷², which is a functional annotator that evaluates the gene content of individual fungi from ITS sequencing data.”*

Line 479-480: For something related to the main conclusion this is well hidden here.

We have now described the agricultural intensity calculation in detail. Ln 495: *“We created a single index of management intensity that could be compared across arable sites and countries, and this index was used in a previous study²⁸. For this, we used the management data obtained*

from the 2017 growing season. The specific management variables that were used in this index were the number of insecticide, herbicide and fungicide applications, the number of tillage events, the maximum tillage depth, and the total amount of mineral nitrogen applied. All of the above parameters were then included in a single management index using principal component analysis and scaling the index between 0 and 1, with 0 indicating the minimum intensity of management practices and 1 indicating the maximum intensity.”

The figures in general (besides fig 1) are unclear and not coherent with the message. Term functional rarity is odd. Further, not all functional groups are equally important (Fig 2C-D; pathogens of pollen, orchid mycorrhizae, undefined symbiotroph, animal endosymbiont) in both ecosystems for their functioning – and many are due to presence of herbivores and/or certain plants in grasslands while they are kept away from the arable fields in purpose. Figure 3 is difficult to understand, same for 4-B.

We believe the revised figures are more coherent. Figures 1 and 2 describe the biogeographical patterns and their drivers. Figures 3 and 4 focus on spatial commonness and rare fungi. We have now described each figure in the caption and clarified details in the Results section.

Reviewers' Comments:

Reviewer #1:

Remarks to the Author:

The authors have significantly improved the manuscript. I think it is now publishable as it is.

Reviewer #2:

Remarks to the Author:

The authors have made substantial changes to the manuscript, and I think it is much improved - particularly in regard to my initial comments. Happy to recommend for publication.

Reviewer #3:

Remarks to the Author:

The manuscript has improved considerably and now shows the general trends and the trends on rare taxa in a balanced manner. Introduction, results and discussion are now well aligned with each other and give justice to this wonderful dataset on fungi. Well done authors! This is a much needed piece of literature, on the patterns governing fungal diversity across Europe.

Specific comments I have left:

The title does not read super well, Maybe better to have Lower soil fungal diversity and fewer rare taxa in arable soils across Europe (the mechanism 'biotic homogenisation' can be added after 'taxa' but I am not sure if it is needed')

Line 136: The types of mycorrhizae could be written neater. Does this also imply ALL of the groups were (relatively) more abundant in grassland or that their COMBINED relative abundance was higher?

Line 160: Suggest to add 'most important' before 'drivers'

Line 170: Please remove 'of'

Line 275-276: to put the latitudes in context (compared to global studies) it would be good to note here what the latitudinal range in Europe is and what is the mid-latitude (I guess 45-50?) with highest diversity?

Line 289: across the latitudinal gradient = across continent

Line 290-292: this is the same as the conclusions on mid-latitudes. This could be tied to that (lines 275-276).

Line 304: add 'both' before 'overall'

Line 443 (and elsewhere): fungal abundances have not been analysed but rather relative abundances of fungal taxa.

Line 493: rather 'predicts' than 'evaluates'

Fig1A legend would help to interpret the figure at one glance!

Fig1D. This figure looks nice but interpretation is quite challenging still. Especially the broad categories of functional groups are not really informative as they can contain anything. The 1C side is more interesting and message is the same?

Is the Fig 4a created with 61+61 sites? Or does it refer only to statistics? It would be good to explain it is the average richness per sample while next figure is cumulative number - would be good to note number of sites there as well (per country).

Suppl. Figure 1: 'Mortierello' not 'Montierello'

Fig S16 is not very convincing pattern, the model has too many 0s to run correctly. Consider omitting. Same with S18, this is very suggestive. In the text it is said that 'diversity' is used but here it is Pcoa axis1 so it is community composition.

REVIEWERS' COMMENTS

Reviewer #1 (Remarks to the Author):

The authors have significantly improved the manuscript. I think it is now publishable as it is.

We thank the reviewer for their positive evaluation.

Reviewer #2 (Remarks to the Author):

The authors have made substantial changes to the manuscript, and I think it is much improved - particularly in regard to my initial comments. Happy to recommend for publication.

We also thank Reviewer 2 for their positive evaluation.

Reviewer #3 (Remarks to the Author):

The manuscript has improved considerably and now shows the general trends and the trends on rare taxa in a balanced manner. Introduction, results and discussion are now well aligned with each other and give justice to this wonderful dataset on fungi. Well done authors! This is a much needed piece of literature, on the patterns governing fungal diversity across Europe.

Thank you very much for your kind words. We appreciate it.

Specific comments I have left:

The title does not read super well, Maybe better to have Lower soil fungal diversity and fewer rare taxa in arable soils across Europe (the mechanism 'biotic homogenisation' can be added after 'taxa' but I am not sure if it is needed')

Many thanks for your suggestion. We have now changed the title to "Biotic homogenization, lower soil fungal diversity and fewer rare taxa in arable soils across Europe".

Line 136: The types of mycorrhizae could be written neater. Does this also imply ALL of the groups were (relatively) more abundant in grassland or that their COMBINED relative abundance was higher?

Members of arbuscular mycorrhizal fungi were significantly higher in the grasslands. We have now clarified it.

Ln 137: "Arbuscular mycorrhizal fungal groups (Glomerallales; **Supplementary Table 2**) showed a higher relative abundance in grasslands than in arable lands, which may be due to the availability of host plants in grasslands."

Line 160: Suggest to add 'most important' before 'drivers'

Added

Line 170: Please remove 'of'

Done

Line 275-276: to put the latitudes in context (compared to global studies) it would be good to note here what the latitudinal range in Europe is and what is the mid-latitude (I guess 45-50?) with highest diversity?

This is a good point. We have added this to contextualize our results.

Line 289: across the latitudinal gradient = across continent

We have changed to "across the continent".

Line 290-292: this is the same as the conclusions on mid-latitudes. This could be tied to that (lines 275-276).

Done.

Line 304: add 'both' before 'overall'

Added

Line 443 (and elsewhere): fungal abundances have not been analysed but rather relative abundances of fungal taxa.

We have added the word "relative".

Line 493: rather 'predicts' than 'evaluates'

Changed

Fig1A legend would help to interpret the figure at one glance!

Legend added

Fig1D. This figure looks nice but interpretation is quite challenging still. Especially the broad categories of functional groups are not really informative as they can contain anything. The 1C side is more interesting and message is the same?

We have added a couple of lines on how to interpret the alluvial diagram. "In these diagrams, various blocks represent clusters, and the streams or flows represent changes in the

composition. For each country, the height of the blocks represents the size of cluster of fungal groups.”

Is the Fig 4a created with 61+61 sites? Or does it refer only to statistics? It would be good to explain it is the average richness per sample while next figure is cumulative number – would be good to note number of sites there as well (per country).

Thank you for your suggestions. We have made these changes in the figure caption.

Suppl. Figure 1: ‘Mortierello’ not ‘Montierello’
Corrected

Fig S16 is not very convincing pattern, the model has too many 0s to run correctly. Consider omitting. Same with S18, this is very suggestive. In the text it is said that ‘diversity’ is used but here it is Pcoa axis1 so it is community composition.

We have deleted Figure S16. If possible, we wish to keep Figure S18 (now S17) since a word of caution is already added in the text. We have changed the axis title to PCoA1 of Rare Fungi.